# Inhibiting acute, axonal DLK palmitoylation is neuroprotective and avoids deleterious effects of cell-wide DLK inhibition

Xiaotian Zhang[1], Heykyeong Jeong[1], Jingwen Niu[1], Sabrina M. Holland[1], Brittany N. Rotanz[1], John Gordon[2], Margret B. Einarson[3], Wayne E. Childers [2] & Gareth M. Thomas [1,4] ✉

Inhibiting dual leucine-zipper kinase (DLK) could potentially ameliorate diverse neuropathological conditions, but a direct inhibitor of DLK's kinase domain caused unintended side effects in human patients, indicative of neuronal cytoskeletal disruption. We sought a more precise intervention and show here that axon-to-soma pro-degenerative signaling requires acute, axonal palmitoylation of DLK. To identify potential modulators of this modification, we screened >28,000 compounds using a high-content imaging readout of DLK's palmitoylation-dependent subcellular localization. Several hits alter DLK localization in non-neuronal cells, reduce DLK retrograde signaling and protect cultured dorsal root ganglion neurons from neurodegeneration. Mechanistically, the two most neuroprotective compounds selectively prevent DLK's stimulus-dependent palmitoylation and subsequent recruitment to axonal vesicles, but do not affect palmitoylation of other axonal proteins assessed and avoid the cytoskeletal disruption associated with direct DLK inhibition. Our hit compounds also reduce pro-degenerative retrograde signaling in vivo, revealing a previously unrecognized neuroprotective strategy.

Dual-leucine zipper kinase (DLK) is an upstream activator (a 'MAP3K') of mitogen-activated protein kinase (MAPK) pathways that is highly expressed in neurons. DLK functions as an evolutionarily conserved stress sensor[1–4] and one major role of DLK is to convey retrograde signals from sites of axonal damage or disruption back to neuronal nuclei[5–10]. Genetic or pharmacological block of DLK inhibits such retrograde signaling, and is neuroprotective, after trophic factor deprivation (TD) in developing neurons, as well as in models of acute injury and of neurodegenerative disease in the mature nervous system[5,6,10–12]. Importantly, DLK was also initially reported to selectively mediate pro-degenerative, but not basal, physiological signaling by downstream MAPKs[5]. These findings suggested that inhibiting DLK could be a promising therapeutic strategy to prevent diverse forms of neurodegeneration and spurred development of inhibitors of DLK's kinase activity[13–15]. Based on these and other encouraging preclinical findings, one such DLK inhibitor was moved forward to a Phase I clinical trial[13,16]. However, numerous patients in this trial developed symptoms indicative of sensory neuropathy which, along with other adverse events, led a significant proportion of those enrolled to reduce or cease dosage[16]. Further analysis revealed that DLK inhibitor treatment elevated levels of neurofilament in patients' plasma, suggestive of axonal cytoskeletal disruption[16]. This effect is likely due to on-target toxicity as plasma neurofilament levels were also elevated when DLK was conditionally and globally knocked out in adult mice[16]. Although it was unclear if this elevation was due to a neuron-intrinsic role of DLK, these findings suggested that broadly inhibiting DLK's kinase activity causes deleterious side effects, cautioning against such a therapeutic approach.

[1]Center for Neural Development and Repair, Lewis Katz School of Medicine at Temple University, Philadelphia, PA 19140, USA. [2]Moulder Center for Drug Discovery, School of Pharmacy, Temple University, Philadelphia, PA, USA. [3]Molecular Therapeutics Program, Fox Chase Cancer Center, Philadelphia, PA, USA. [4]Department of Neural Sciences, Lewis Katz School of Medicine at Temple University, Philadelphia, PA 19140, USA. ✉e-mail: gareth.thomas@temple.edu

We therefore considered the possibility that more refined strategies to inhibit specific pools of DLK, particularly those involved in pro-degenerative retrograde signaling, might afford neuroprotection without cytoskeletal disruption. In particular, our prior work revealed that DLK is covalently modified with the lipid palmitate[8]. This modification, palmitoylation, targets DLK to lipid membranes and in neurons is critical for DLK to hitchhike on axonal trafficking vesicles[8]. Palmitoylation is also critical for DLK's function; by using shRNA knockdown and rescue to replace endogenous DLK with a palmitoyl-site mutant, we found that palmitoylation is essential for DLK-dependent retrograde signaling and for subsequent neurodegeneration, both in cultured neurons and in vivo.[8,9,17]. Importantly, the protective effects of DLK palmitoyl-site mutation are as robust as complete loss of DLK[9,17].

Genetically preventing DLK palmitoylation (e.g. by mutating DLK's palmitoylation site at the genomic level, perhaps using Clustered Regularly Interspaced Short Palindromic Repeats (CRISPR)-based methods) is unlikely to be feasible therapeutically. However, the above findings suggested that pharmacologic block of palmitoyl-DLK-dependent signaling might serve as an alternate neuroprotective strategy. Building on this notion, we previously developed a high content imaging screen to identify compounds that alter the palmitoylation-dependent localization of transfected, GFP-tagged DLK (DLK-GFP) in non-neuronal human embryonic kidney (HEK) 293 T cells[18]. Wild type (wt) DLK-GFP normally localizes to Golgi membranes and small vesicle-like puncta in these cells, but pharmacologic or genetic block of DLK palmitoylation results in a diffuse DLK-GFP signal[8,18]. DLK-GFP's punctate localization can thus be used as a proxy for DLK palmitoylation that is compatible with a High Content Screening (HCS)-based approach[18]. A pilot screen using this assay identified a compound, ketoconazole, which reduced DLK-GFP's punctate localization in HEK293T cells and also partially reduced acute DLK-dependent signaling in neurons shortly after TD[18]. However, we reasoned that this partial, acute inhibition might not be sufficient to protect DRG neurons against degeneration induced by prolonged TD. We, therefore, sought to expand our screen using larger libraries, with the objective of identifying more potent compounds capable of preventing not just DLK-dependent signaling but also DLK-dependent neurodegeneration itself.

In this work, we identify 33 compounds (out of a primary screen of >28,000 compounds) that dose-dependently reduce DLK-GFP's palmitoylation-dependent localization in non-neuronal cells. Eight of these compounds inhibit pro-degenerative DLK-dependent signaling in neurons and a further subset protects DRG neuronal cell bodies and axons against degeneration induced by prolonged TD. Mechanistically, TD induces an acute palmitoylation-dependent change in the axonal localization of DLK that is prevented by our most neuroprotective hits, providing a plausible explanation for their neuroprotective activity. Our top hits do not affect palmitoylation of other axonal proteins tested, do not affect other molecular events induced by TD, and do not affect basal levels of DLK palmitoylation or signaling. Crucially, and consistent with this latter finding, our top hits also do not phenocopy the disruption of the axonal cytoskeleton seen after DLK kinase inhibition. Moreover, intravitreal injection of our top hits significantly reduces DLK-dependent pro-degenerative retrograde signaling in the retina. These new classes of inhibitors thus reduce DLK-dependent pro-degenerative retrograde signaling without causing side effects associated with global DLK inhibition.

## Results

### A neuron-intrinsic role of DLK in governing cytoskeletal integrity in healthy axons

Treatment of human patients with the DLK kinase domain inhibitor GDC-0134, and cKO of DLK (gene name *Map3k12*) in mice, both cause elevated plasma neurofilament levels, a biomarker of cytoskeletal disruption and neurodegeneration[19]. This finding suggests that DLK activity is required for normal structure and function of the axonal cytoskeleton. The association between GDC-0134 treatment and sensory neuropathy[16], a condition often linked to pharmacological or genetic disruption of the cytoskeleton of dorsal root ganglion (DRG) sensory neurons[20–23], further supports this conclusion.

These phenotypes could be due to roles of DLK in a different subcellular compartment, or even in a different cell type, but we hypothesized that DLK might be intrinsically required for axonal cytoskeletal integrity in DRG neurons. To test this possibility, we assessed the effect of acute DLK inhibition on the distribution of Neurofilament heavy chain (NF-200) and neuron-specific βIII tubulin (Tuj1). In healthy DRG neurons treated with vehicle, NF-200 and βIII tubulin were smoothly and uniformly distributed along axons. However, in cultures treated with the widely used DLK inhibitor GNE-3511 (structurally similar to GDC-0134[12,16]), NF-200 and Tuj1 signals were no longer smoothly distributed along axons but instead partly accumulated in axonal distortions (Fig. 1A). Moreover, the vesicle marker VAMP2 also accumulated in these structures (Fig. 1B), suggesting that the cytoskeletal disruption caused by DLK inhibition in turn, or in addition, disrupted axonal transport. DLK itself was also enriched in these accumulations (Fig. 1A, B), suggesting that blockade of endogenous DLK activity at these sites is likely responsible for the observed distortions and accumulations. Quantified data for the distribution of NF-200, Tuj1, DLK, and VAMP2 are shown in Fig. 1C−F, respectively, and examples of image processing and analysis methods used to generate the quantified data are in Fig S1. Collectively, these findings uncover a previously unrecognized role for DLK in maintaining the integrity of axonal neurofilaments, microtubules, and/or vesicle-based transport in healthy axons.

### Palmitoylation-dependent recruitment of DLK to axonal vesicles is implicated in axonal retrograde signaling

These findings increase the likelihood that the peripheral neuropathy symptoms caused by DLK inhibition are due to dysregulated cytoskeletal integrity and/or axonal transport in DRG sensory neurons themselves. We therefore considered ways to specifically prevent pro-degenerative DLK signaling without globally inhibiting DLK. We previously reported that DLK must be palmitoylated to support axonal retrograde signaling[8,9,17]. In addition, a subsequent study reported that TD, which triggers pro-degenerative DLK signaling in developing DRG neurons[5], acutely increases DLK palmitoylation, and also induces DLK recruitment to axonal vesicles[24]. However, the functional role of this acute, TD-induced change in DLK localization was not addressed. It was also unclear whether this change of DLK localization requires, or simply correlates with, increased DLK palmitoylation. As a first step to address these questions, we infected DRG neurons to express wild type, myc-tagged DLK (wtDLK-myc), subjected neurons to TD and then immunostained 3 h later to detect myc signal. TD increased wtDLK-myc recruitment to axonal puncta (Fig. 2A, B), which, based on a prior study assessing the same timepoint (3 h post-TD) are likely vesicles[8,24].

We also investigated whether DLK recruitment to axonal vesicles involves acute palmitoylation of DLK itself. To address this question, we used spot cultures, in which DRG axons extend from centrally plated cell bodies, thus allowing specific harvesting of axonal material. We subjected spot culture axonal fractions to a biochemical palmitoylation assay, acyl-biotin exchange (ABE) in which thioester-linked acyl groups (most commonly palmitate) are exchanged for biotin, allowing capture of the resultant biotinyl-proteins on avidin beads. Western blots of ABE fractions from DRG axons, obtained 2 h post-TD, confirmed that TD acutely increases DLK palmitoylation (purity of axonal preparations for experiments of this type is confirmed in Fig. S2). A broad spectrum inhibitor of cellular palmitoylation, 2-bromopalmitate (2BP;[25]) prevented both the TD-induced increase in DLK palmitoylation, and DLK recruitment to vesicles, increasing the

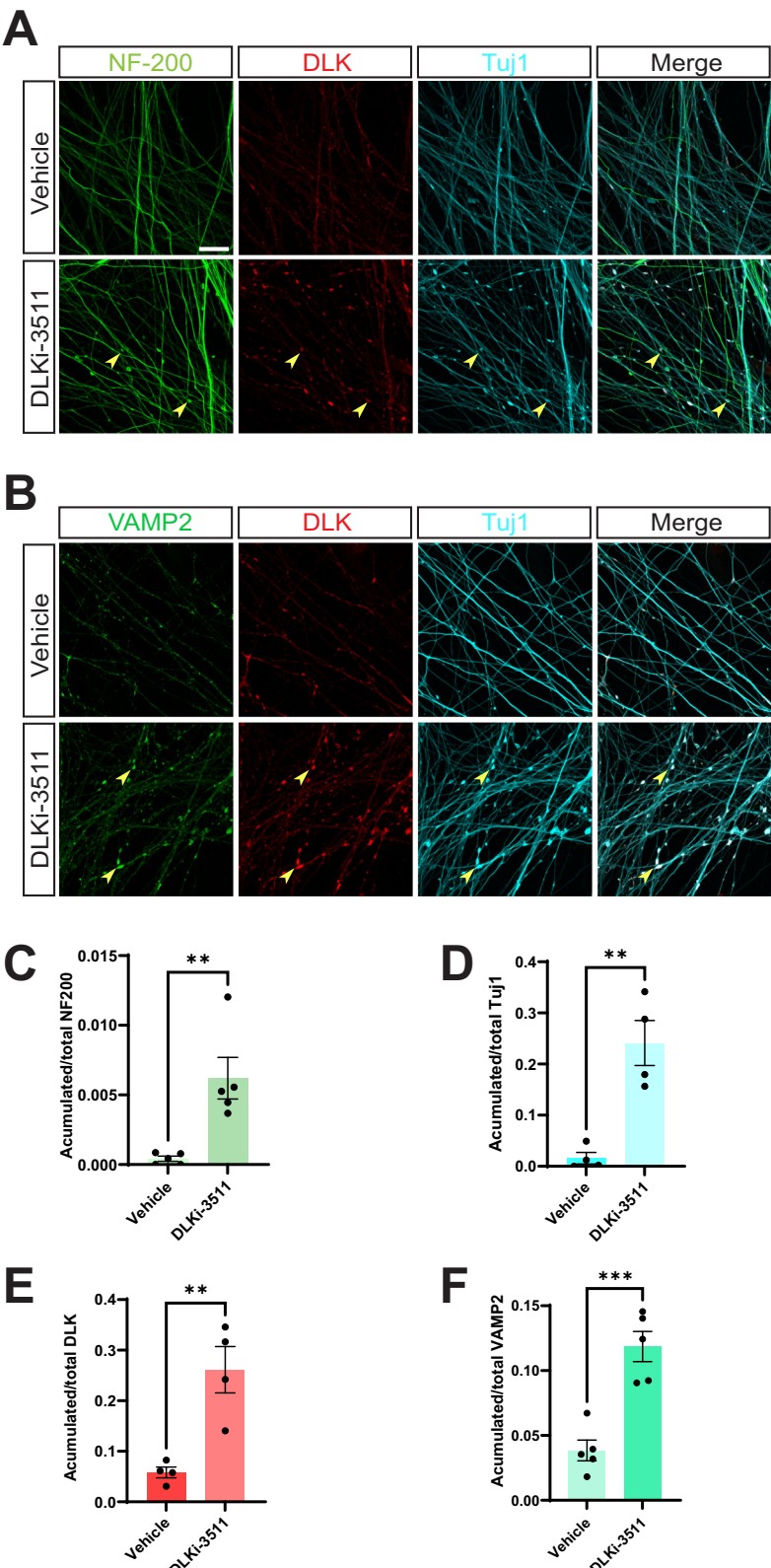

likelihood that the former process drives the latter (Fig. 2A–D). These findings confirm that TD triggers acute, axonal palmitoylation of DLK, and suggest that this acute palmitoylation is required for DLK to associate with axonal vesicles.

To assess the functional importance of axonal palmitoylation for DLK-dependent retrograde signaling, we plated DRG neurons in microfluidic chambers and selectively withdrew trophic support from distal axonal compartments that had been acutely treated with either 2BP or vehicle. Selective, acute inhibition of axonal palmitoylation with 2BP prevented TD-induced c-Jun phosphorylation in DRG cell bodies (Fig. 2E, F). This finding is consistent with a model in which acute, axonal palmitoylation of DLK is critical for TD-induced, pro-

**Fig. 1 | DLK kinase domain inhibition disrupts axonal integrity in DRG neurons.**
**A** Images of cultured DRG neurons that had been treated with DMSO vehicle or
500 nM DLK inhibitor GNE-3511 (DLKi-3511) for 4 hours and fixed and stained with
the indicated antibodies. **B** As A, except that neurons were fixed to detect vesicle
marker VAMP2 rather than NF-200. **C–F** Quantified data from A and B confirm that
DLK inhibition disrupts distribution of NF-200 (neurofilament heavy chain; **1C**; 5
independent cultures per condition) and Tuj1 (neuron-specific tubulin; **1D**; 4
independent cultures per condition) and leads to accumulations of DLK (**1E**, 4

independent cultures per condition) and VAMP2 (**1F**, 5 independent cultures per
condition) in axons, suggestive of cytoskeletal dysregulation and/or impaired
vesicle-based transport. Yellow arrowheads highlight examples of disruption/
accumulation of the respective signals. Unpaired two-sided t tests reveal significant
effects of DLKi-3511 vs. Vehicle treated conditions for NF-200 signal ($p = 0.005$),
DLK signal ($p = 0.005$), Tuj1 signal ($p = 0.0025$), and VAMP2 signal ($p = 0.0005$).
Scale bar: 20 μm (all panels). Data are presented as mean values +/− SEM. Source
data are provided in the Source Data file.

degenerative retrograde signaling. Importantly, these results also
suggested that selectively preventing acute, palmitoylation-
dependent recruitment of DLK to vesicles could be a neuroprotec-
tive strategy that might circumvent the side effects associated with
inhibiting all cellular pools of DLK.

## High content screening to identify modulators of DLK's palmitoylation-dependent localization

To identify compounds that might specifically prevent palmitoylation-
dependent retrograde signaling by DLK, we expanded a high content
imaging screen designed to identify small molecules that alter the
palmitoylation-dependent localization of transfected DLK-GFP in
HEK293T cells (Fig. 3A[18]). Our expanded screen consisted of 28,400
compounds from the Maybridge and Enamine diversity sets. The
properties of compounds that make up these libraries are fully
described under Methods and the reproducibility of the screen itself,
(Z-factor - initially described in ref. [18]) is shown in Fig. S3A, B for the
scaled-up conditions used for this specific set of compounds. Of these
28,400 compounds, 1,723 reduced the number of cells that detectably
expressed a cotransfected nuclear marker, mCherry tagged with a
Nuclear Localization Sequence (mCh-NLS), by >30%. We previously
used this cut-off to filter out compounds that may broadly reduce
transcription and/or translation[18], so these 1723 compounds were not
pursued further. For the remaining 26677 compounds, we calculated
the number of DLK-GFP puncta per mCh-NLS-expressing cell (puncta
per NLS; P/NLS) and the average intensity of those puncta (Vesicle
Average Intensity; VAI). 33 compounds increased P/NLS by >1.8-fold
and were also not pursued further. Of the remaining 26644 com-
pounds, 375 compounds reduced both P/NLS and VAI by >2 SD below
the mean of all determinations (Fig. 3B, C) and were selected for
follow-up analysis.

The 375 compounds were then re-assayed in triplicate at 3 dilu-
tions (10 μM, 3 μM, 1 μM) to confirm their effects on P/NLS and VAI
and to assess dose-dependence. 33 compounds were effective and
showed dose-dependence in one or both readouts in this follow-up
assay (Table S1). Fresh stocks of these 33 compounds (with positions
in the screen shown in Fig. S4 and chemical structures shown in
Fig. S5) were purchased for follow-up neuronal assays, in particular to
assess their ability to inhibit TD-induced palmitoyl-DLK-dependent
signaling.

## Multiple hit compounds inhibit DLK-dependent c-Jun phos-phorylation in trophic factor-deprived DRG neurons

We next asked whether compounds identified in our HEK293T
cell screen could inhibit TD-induced DLK signaling in DRG neurons.
Following TD, DLK signals via c-Jun N-terminal kinases (JNKs), which
in turn phosphorylate transcription factors including c-Jun. Phos-
phorylation of c-Jun (p-c-Jun) is thus a widely-used readout to
assess the somatic response to pro-degenerative stimuli including,
but not limited to, TD[5,8–10,17,26]. Consistent with prior work, a DLK
kinase domain inhibitor (DLKi, GNE-3511) completely blocked
p-c-Jun signals induced by TD[17]. Many of our hit compounds also sig-
nificantly inhibited TD-induced c-Jun phosphorylation (Fig. 4A, B),
suggesting that these compounds not only disrupt DLK localization in
non-neuronal cells but also attenuate DLK-dependent signaling in
neurons.

## A subset of hit compounds protects DRG cell bodies and axons from the effects of prolonged trophic factor deprivation

We further investigated whether hit compounds that inhibit TD-
induced p-c-Jun signaling could also confer protection against
degeneration in DRG neurons subjected to prolonged TD. We used two
assays to assess the effects of prolonged TD; a loss of staining with the
vital dye Calcein-AM to assess cell body viability, and an Interactive
Watershed-based analysis of phase contrast images to assess loss of
axon integrity. In the former readout, TD greatly decreased the num-
ber of Calcein-AM positive neuronal cell bodies that costained with the
DNA dye Hoechst 33342 i.e. viable cell bodies (Fig. 5A, B). GNE-3511 and
two of our hit compounds, **8** and **13**, prevented TD-induced loss of
Calcein-AM-positive neurons (Fig. 5A, B). Despite its broad spectrum of
action, 2BP also prevented TD-induced loss of Calcein-AM-positive
neurons, while two other compounds, **6** and **30**, partially prevented
this loss (Fig. 5A, B).

In parallel, we assessed axon integrity in phase contrast images of
the same neurons. Because axons were not fully degenerated at this
time point (chosen due to optimal dynamic range for the Calcein-AM/
Hoechst assay) we did not use an axon degeneration index commonly
used to quantify fully fragmented axons[27], but instead used the ImageJ/
Fiji Interactive Watershed function[28] to identify long sections of con-
tinuous (i.e. unblebbed, phenotypically intact) axons. GNE-3511, 2BP, **6**,
**8**, and **13** all protected axon integrity as assessed using this assay
(Fig. 5A, C). Due to their ability to protect both neuronal cell bodies and
axons to a similar extent as GNE-3511, **8** and **13** were selected for further
analysis. Examples of image processing steps for the axon integrity
analysis are shown in Fig S6.

We then performed a follow-up assay to confirm that **8** and **13**
indeed effectively reduce DLK-GFP punctate localization in transfected
non-neuronal cells (Fig. S7). Consistent with results of our primary
screens ([18] and Fig. 3), **8, 13** and ketoconazole (the top hit from our
pilot screen[18]) all reduced DLK-GFP punctate localization, although
none did so as effectively as 2BP (Fig. S7). GNE-3511 reduced DLK-GFP
punctate signal in some cells but caused large, bright accumulations of
DLK-GFP in others (Fig. S7), an unexpected effect that is an interesting
area for future study. A full description of our screening procedure,
with validation assays and inclusion/exclusion criteria for specific
steps, is in Fig. S8, with additional details in Table S2.

We also assessed the dose-dependence of neuronal cell body
protection by **8** and **13**. Both compounds showed essentially a maximal
effect at the 10 μM dose used in our primary assays, justifying use of
this concentration in our initial and follow-up assays (example images
in Fig. S9A, C, quantified in Fig. S9B, D). $IC_{50}$ values for both **8** and **13**
were in the low micromolar range for the neuronal cell body protec-
tion assay (Fig. S9B, D).

## Neuroprotective hit compounds selectively prevent palmitoylation-dependent DLK localization and signaling after TD

Given that **8** and **13** are neuroprotective and were identified in a screen
assessing DLK's palmitoylation-dependent localization, we assessed
the effect of these compounds on acute, TD-induced recruitment of
DLK to axonal vesicles, which is palmitoylation-dependent (Fig. 2).
Both **8** and **13** prevented TD-induced recruitment of DLK-myc to
axonal vesicles (Fig. 6A, B). Although 2BP also prevented TD-induced

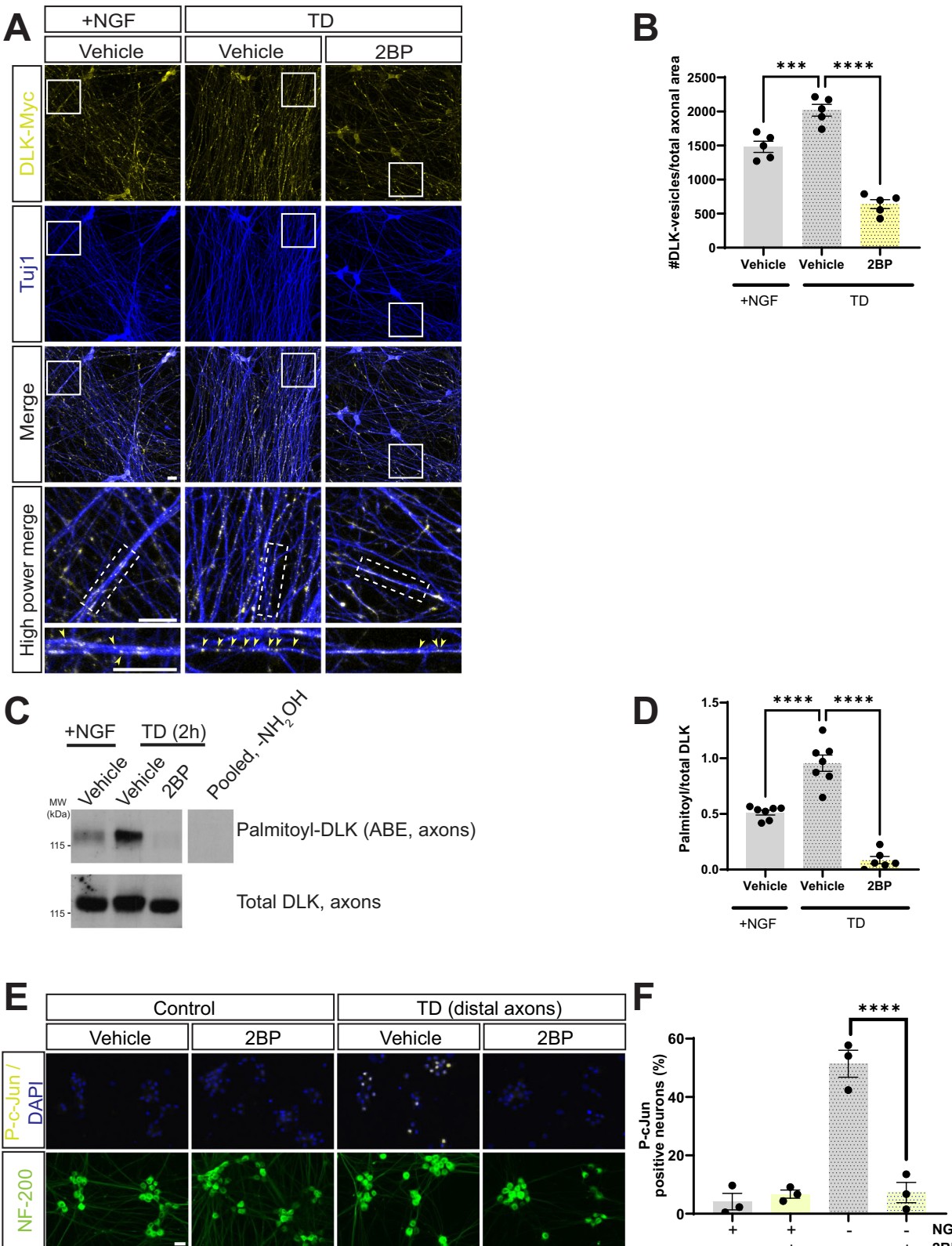

recruitment of DLK to axonal vesicles, it also reduced the number of DLK-positive vesicles in healthy axons. In contrast, **8** and **13** did not affect localization of DLK to vesicles in healthy axons (Fig. 6A, B). In addition, ABE assays from DRG neuron spot culture distal axons revealed that **8** and **13** also prevented the TD-induced increase in DLK palmitoylation (Fig. 6C, D) but that neither compound decreased DLK palmitoylation in healthy cultures (Fig. S10). 2BP treatment of healthy

cultures showed a trend towards decreased DLK palmitoylation in axon fractions but this did not reach statistical significance. These results suggest that **8** and **13** selectively inhibit TD-induced, but not basal, DLK palmitoylation.

The effects of **8** and **13** on DLK's axonal palmitoylation and localization suggested that these compounds might also prevent TD-induced signaling by DLK. Indeed, when we assessed DLK's direct

**Fig. 2 | Trophic deprivation-induced recruitment of DLK to axonal vesicles and DLK-dependent retrograde signaling are both palmitoylation-dependent.**
**A** Cultured DRG neurons were lentivirally infected to express myc-tagged wild type DLK (wtDLK-myc) and left untreated (+NGF) or were subjected to trophic factor deprivation (TD) in the presence of 2-bromopalmitate (2BP) or DMSO vehicle. Cultures were fixed 3 h post-TD and immunostained with the indicated antibodies. The bottom two rows of images (*High Power Merge*) show magnified views of the boxed region in the image directly above. Arrowheads in bottom row panels indicate DLK-myc vesicle-like puncta. Scale bars, 20 μm (all panels). **B** Quantified data from *A* confirm that the TD-induced increase in axonal DLK puncta (presumptive vesicles[24]) is prevented by 2BP i.e. is palmitoylation-dependent. ***; $p = 0.0008$; ****; $p < 0.0001$, two-sided one-way ANOVA, Dunnett's post hoc test. 5 independent cultures per condition. **C** Western blots of ABE (palmitoyl-) fractions of axonal lysates from DRG neuron spot cultures that had been treated as indicated prior to

lysis. The righthand lane is a side-by-side exposure from a parallel control sample omitting the key ABE reagent NH$_2$OH, run on the same gel, with intervening spacer lanes cropped. **D** Quantified data from *C* confirm that TD increases axonal DLK palmitoylation, which is prevented by 2BP. ****; $p < 0.0001$, two-sided one-way ANOVA, Dunnett's post hoc test. 7 independent cultures per condition (+NGF/ Vehicle, TD/Vehicle); 6 independent cultures (TD/2BP). **E** Images of cell body chambers of DRG microfluidic cultures, fixed and stained with the indicated antibodies 4 h after selective treatment of distal axonal compartments as indicated. Scale bars, 50 μm. **F** Quantified data from *E* confirm that TD-induced retrograde signaling requires acute palmitoylation in distal axons. ****; $p < 0.0001$, two-sided one-way ANOVA, Dunnett's post hoc test. 3 independent cultures per condition. Data are presented as mean values +/− SEM. Source data are provided in the Source Data file.

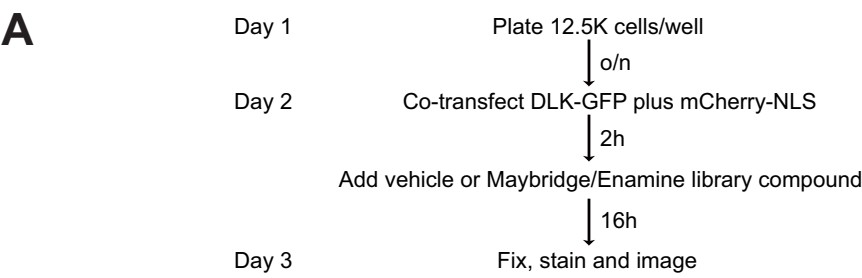

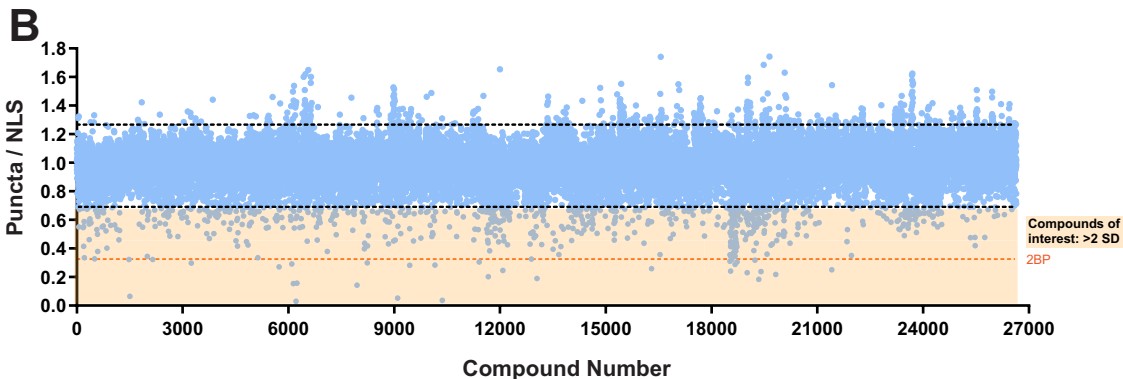

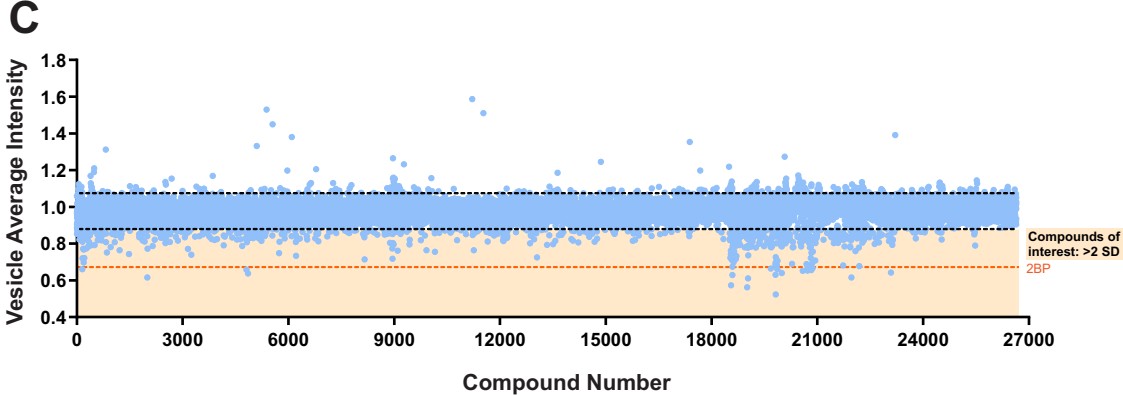

**Fig. 3 | An expanded High Content Imaging screen identifies compounds that inhibit the punctate localization of DLK-GFP. A** Experimental design of the high-throughput screen to identify compounds that inhibit DLK-GFP punctate localization, adapted from[18]. **B, C** Evaluation of the effect of 26,677 compounds that passed cut-offs (of 28,400 compounds screened) from the Maybridge and Enamine Libraries™ on DLK-GFP puncta per transfected cell (Puncta/NLS) and average brightness of those puncta (Vesicle Average Intensity). Black dotted lines indicate

2 standard deviations (2 SD) above and below the mean of all determinations that passed cut-offs. Compounds that reduced both readouts by >2 SD (region highlighted in pale orange) were selected for further analysis. Dark orange dotted lines in **B** and **C** indicate the average reduction seen in each readout with 2-Bromopalmitate (2BP, tool compound positive control). A single biological replicate was run for each compound in this primary screen. Source data are provided in the Source Data file.

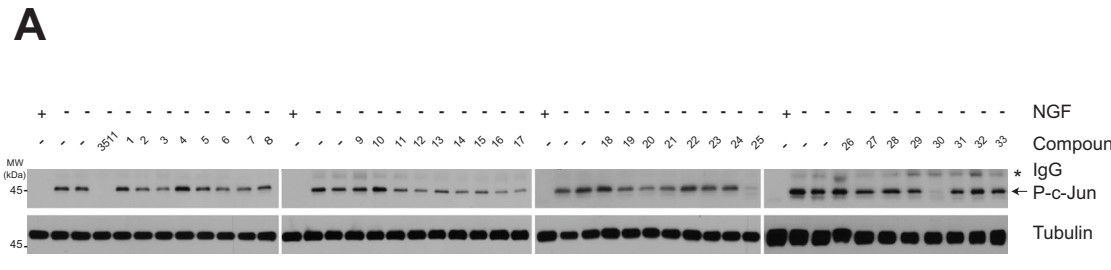

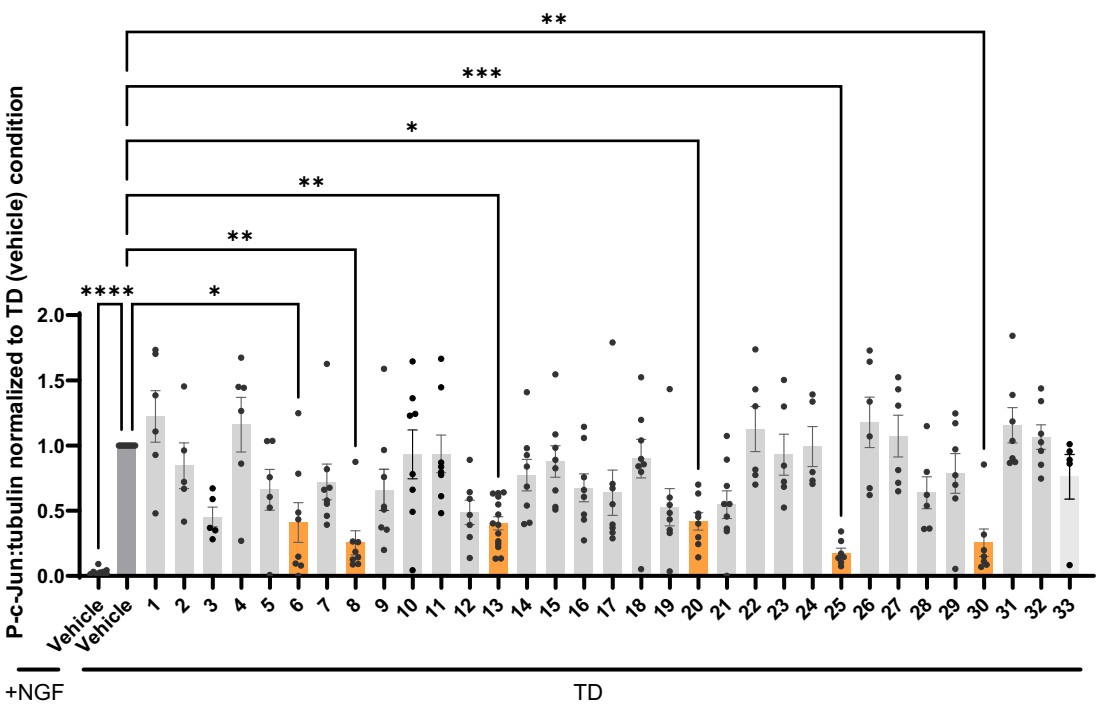

**Fig. 4 | Multiple hit compounds reduce TD-induced c-Jun phosphorylation.**
**A** Western blots of lysates from cultured DRG neurons that had been treated with vehicle (DMSO), GNE-3511 (3511) or the indicated hit compounds for 1 h at 5 days in vitro (DIV 5), followed by 2.5 h TD in the continued presence of the indicated compounds, or that had been left unstimulated (+ NGF). The secondary antibody used on the p-cJun blot also weakly recognizes residual anti-NGF IgG used during TD (indicated by asterisk). Subsets of compounds were assayed side-by-side, but in batches indicated by spaces between individual blots. **B** Quantified data from *A*, of p-cJun:tubulin, normalized to TD (vehicle) condition. Compounds whose effect on p-c-Jun:tubulin differed significantly from TD (vehicle) condition are highlighted

with orange bars. Statistical significance versus TD (vehicle) control was as follows: +NGF (vehicle): $p < 0.0001$; **6**: $p = 0.0303$; **8**: $p = 0.0014$; **13**: $p = 0.0033$; **20**: $p = 0.0309$ **25**: $p = 0.0005$; **30**: $p = 0.0029$, two-sided Kruskal Wallis tests with Dunn's post hoc test for multiple comparisons. Data are plotted for the following number of independent cultures per condition: +NGF/Vehicle: 9; TD/Vehicle: 10; TD/**1**: 6; TD/**1**: 6; TD/**2**: 5; TD/**3**: 5; TD/**4**: 6; TD/**5**: 6; TD/**6**: 8; TD/**7**: 8; TD/**8**: 8; TD/**9**: 8; TD/**10**: 8; TD/**11**: 8; TD/**12**: 7; TD/**13**: 14; TD/**14**: 8; TD/**15**: 8; TD/**16**: 8; TD/**17**: 8; TD/**19**: 8; TD/**20**: 8; TD/**21**: 9; TD/**22**: 6; TD/**23**: 6; TD/**24**: 5; TD/**25**: 7; TD/**26**: 6; TD/**27**: 6; TD/**28**: 6; TD/**29**: 7; TD/**30**: 7; TD/**31**: 7; TD/**32**: 7; TD/**33**: 7. Data are presented as mean values +/- SEM. Source data are provided in the Source Data file.

substrate, MKK4, we found that **8** and **13** both prevented TD-induced increases in MKK4 phosphorylation (pMKK4) (Fig. 6E, F). In a prior study, 2BP also blocked TD-induced increases in pMKK4, consistent with a model in which this event requires DLK and it palmitoylation[17]. In contrast, but consistent with their lack of effect on basal DLK palmitoylation and localization, neither **8**, **13** or 2BP reduced basal pMKK4 levels in healthy axons (Fig. S11).

**Neuroprotective hit compounds do not reduce palmitoylation of other axonal palmitoyl-proteins assessed and do not affect other aspects of TD signaling**

We next sought to more rigorously assess the specificity of **8** and **13**'s action towards palmitoyl-DLK versus other palmitoyl-proteins and

cellular events in trophically deprived axons. We found that neither **8** nor **13** affected palmitoylation of three well known axonal palmitoyl-proteins that we assessed: the abundant axonal palmitoyl-protein growth-cone associated protein GAP-43 (Fig. S12A, B), the alpha sub-unit of the G protein $G_o$ ($G\alpha_o$, Fig. S12C, D), and the axonal protein acyltransferase ZDHHC5[29–32] (Fig. S12E, F). We also assessed the effect of **8** and **13** on TD-induced shutdown of the ERK and Akt signaling pathways, two well-described events triggered by TD[5,17,33]. Similar to our prior results with 2BP[17], neither **8** nor **13** prevented TD-induced dephosphorylation/inactivation of ERK and Akt (Fig. S13A–D). This latter finding is consistent with a model in which the neuroprotective ability of these compounds is due to their selective block of TD-induced DLK palmitoylation.

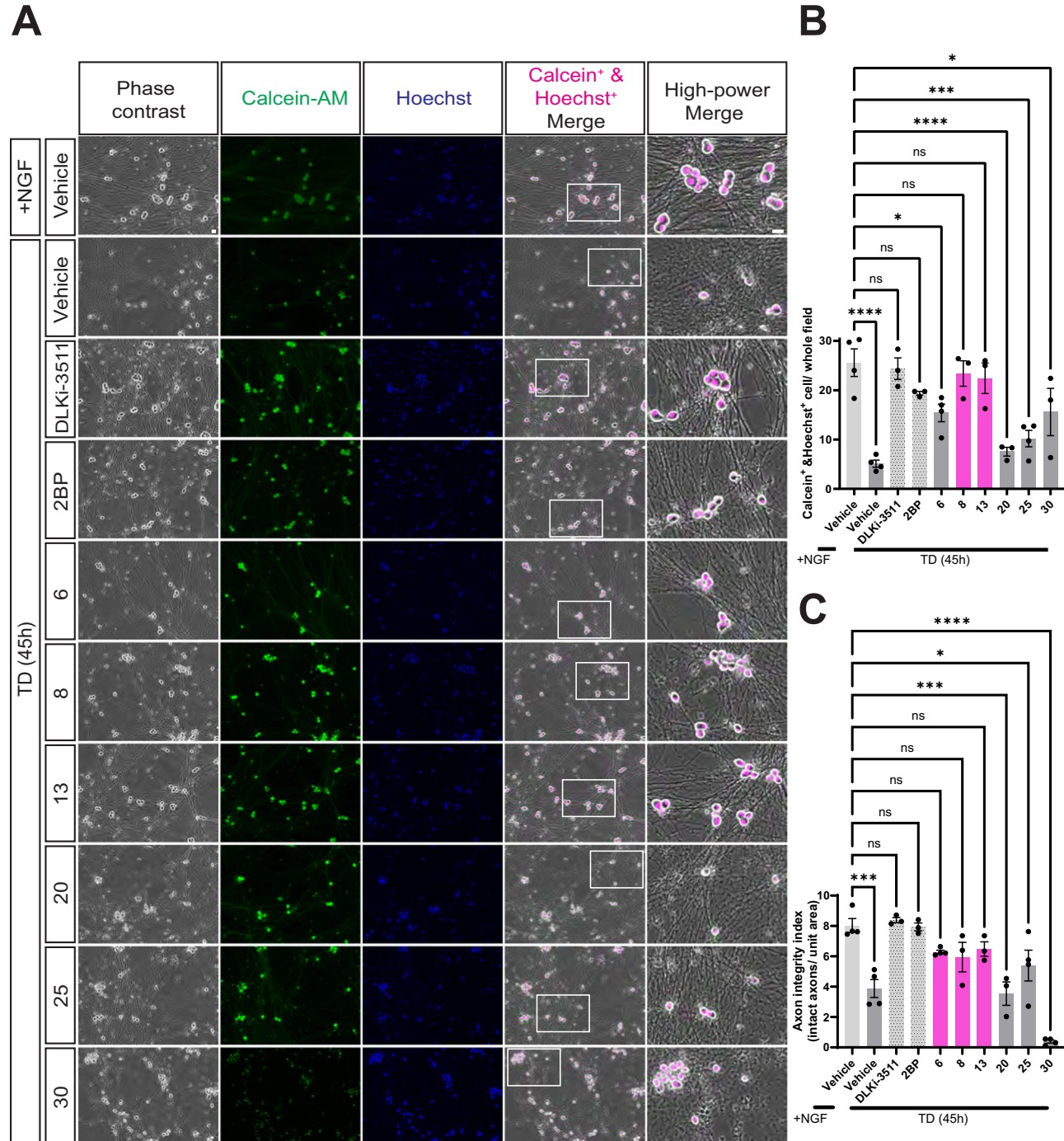

## Neuroprotective hit compounds differentially affect other forms of DLK-dependent signaling but do not cause cytoskeletal disruption or vesicle accumulation

We next asked if **8** and **13** block other forms of DLK-dependent neuronal signaling. Both **8** and **13** prevented the increase in c-Jun phosphorylation that occurs after prolonged (16 h) treatment with the microtubule disrupting agent nocodazole (Fig. S14A, B). This event was previously attributed to activation of the DLK pathway[34,35] and, consistent with a prior report[35], nocodazole-induced c-Jun phosphorylation was also prevented by the DLK kinase domain inhibitor GNE-3511 (Fig. S14A, B).

We also assessed axotomy-induced c-Jun phosphorylation in DRG neurons, which we and others have previously reported to be DLK-dependent, in studies that used immunostaining as a readout[8,36]. Here

we assessed cJun phosphorylation using western blotting to match our other assays (e.g. those shown in Fig. 4, Fig. S14A). Experiments with an antibody that recognizes cJun phosphorylated on Ser63 (Ser63(P); a known JNK phosphorylation site that we and others have routinely used as a DLK-dependent readout[5,6,8,17,37], suggested that **8** prevented axotomy-induced c-Jun phosphorylation at Ser63, but **13** did not (Fig. S14C, D). Because the anti-c-Jun Ser 63(P) antibody recognized a ladder of bands on western blots of cell body fractions of axotomized lysates (Fig. S14C), we also assessed axotomy-induced phosphorylation of c-Jun at Ser73, another site directly phosphorylated by JNK[38]. The anti-c-Jun Ser 73(P) antibody recognized a single band on western blots, axotomy-induced phosphorylation of which was again prevented by **8** but not by **13** (Fig. S14C, E). GNE-3511 prevented axotomy-induced phosphorylation of both sites, consistent with the regulation

**Fig. 5 | A subset of hit compounds reduces TD-induced neurodegeneration.**
**A** Phase contrast images (first column), Calcein-AM signal (2nd column) and Hoechst 33342 DNA signal (3rd column) of DRG neurons that were maintained in NGF (+ NGF) or subjected to TD for 45 h in the presence of vehicle (DMSO) or the indicated hit compounds that significantly reduced TD-induced c-Jun phosphorylation in Fig. 4. The fourth column shows an overlay of the phase contrast image with the Calcein/Hoechst double-positive signal (latter false-colored in magenta). The fifth column shows magnified views of the boxed region in the corresponding fourth column. Scale bars, 20 μm (all panels). **B** Number of Calcein/Hoechst double-positive cells per field (i.e. viable cell bodies), quantified from images from **A**. Magenta shaded bars and *ns* labels indicate hit compounds whose effect on cell body viability post-TD does not differ significantly from the +NGF (vehicle) control condition. GNE-3511 and 2BP also protected cell bodies from the effects of TD. The slight increase in Calcein-AM signal in cultures subjected to TD is consistent with a prior report[55]. Statistical significance versus +NGF (vehicle) control was as follows: **6**: $p = 0.0187$; **20**: $p < 0.0001$; **25**: $p = 0.0003$; **30**: $p = 0.0375$. TD (vehicle) condition

also differed significantly from +NGF (vehicle) $p < 0.0001$. ns: not significant, two-sided one-way ANOVA, Dunnett's post hoc test. Data are plotted for the following number of independent cultures per condition: +NGF/Vehicle: 4; TD/Vehicle: 4; TD/DLKi-3511: 3; TD/2BP: 3; TD/**6**: 4; TD/**8**: 3; TD/**13**: 3; TD/**20**: 3; TD/**25**: 4; TD/**30**: 3. **C** Axon integrity index, determined by counting the number of continuous elongated structures (i.e. unbroken axons) per unit area, quantified from phase contrast images (first column) in *A*. Magenta shaded bars and *ns* labels indicate hit compounds for which the axon integrity index post-TD did not differ significantly from the +NGF (vehicle) control condition. GNE-3511 and 2BP also protected axons from the effects of TD. Statistical significance versus +NGF (vehicle) control was as follows: **20**: $p = 0.0001$; **25**: $p = 0.017$; **30**: $p < 0.0001$. TD (vehicle) condition also differed significantly from +NGF (vehicle) $p = 0.0001$. ns: not significant, two-sided one-way ANOVA, Dunnett's post hoc test. Data are plotted for the following number of independent cultures per condition: +NGF/Vehicle: 4; TD/Vehicle: 4; TD/DLKi-3511: 3; TD/2BP: 3; TD/**6**: 4; TD/**8**: 3 TD/**13**: 3; TD/**20**: 3; TD/**25**: 4; TD/**30**: 3. Data are presented as mean values +/- SEM. Source data are provided in the Source Data file.

of both sites by the DLK/JNK pathway (Fig. S14C–E). GNE-3511 also reduced c-Jun-S73(P) below basal levels, whereas **8** selectively prevented the axotomy-induced phosphorylation at this site (Fig. S14C). Taken together, these findings suggest that **8** blocks multiple forms of DLK/JNK-dependent signaling, whereas **13** effectively blocks DLK/JNK signals induced by certain stimuli (TD, nocodazole) but is ineffective against others (axotomy).

As a final assessment of their effects in cultured neurons, we asked whether **8** and/or **13** phenocopy the cytoskeletal disruption and vesicle aggregation seen with GNE-3511. In contrast to this DLK kinase domain inhibitor, neither **8** nor **13** caused mislocalization of axonal NF-200, Tuj1, DLK and VAMP2 (Fig. 7A–F). Together, these findings suggest that **8** and **13** selectively inhibit pro-degenerative retrograde axonal signaling by DLK but minimally affect roles of DLK in healthy axons.

### Protective compounds identified in primary neurons blunt prodegenerative retrograde signaling in vivo

Finally, we sought to assess the translational potential of **8** and **13**. We focused on the ability of these compounds to reduce pro-degenerative retrograde signaling to RGC cell bodies after optic nerve crush (ONC) in vivo, a process that is highly dependent on (palmitoyl)-DLK[6,7,9,39]. Although MDCK-MDR1 assays suggested that both **8** and **13** can likely cross the blood-brain barrier (BBB), neither **8** nor **13** was stable in liver microsomes in the presence of NADPH (Table S3) suggesting that stability of these compounds in vivo might be low if they were to be systemically delivered. While the ability of **8** to inhibit two of three cytochrome P450 enzymes tested might suggest additional stability in vivo, the possibility that this property might induce other systemic effects, and the overall instability of **13** led us to pursue a different approach. In particular, we reasoned that **8** and **13** might still be active if delivered locally to the eye via intravitreal injection. Because intravitreally injected compounds can rapidly drain from the eye[40,41], we isolated retinas 15 h post-ONC. At this time point, we observed robust, ONC-induced c-Jun phosphorylation in RGCs in retinas from mice that had been intravitreally injected with DMSO vehicle immediately post-ONC (Fig. 8A, B). Importantly, intravitreal injection of either **8** or **13** reduced DLK-dependent responses of RGCs (ONC-induced c-Jun phosphorylation) to a similar extent as GNE-3511 (Fig. 8A, B). RGCs were identified with the marker Brn3A, whose expression remains constant at this time post-ONC;[42,43] Fig. 8A). These findings suggest that **8** and **13** inhibit DLK-dependent axon-to-soma signaling in vivo, revealing a potential therapeutic approach that may more selectively block pro-degenerative retrograde signaling by DLK without the side effects of globally inhibiting all pools of this kinase.

### Discussion

Given the numerous preclinical studies that identified DLK as a promising therapeutic target[6,7,11,12,44], the number of patients who

experienced sensory neuropathy and other adverse events in a recent clinical trial of a DLK kinase domain inhibitor was disappointing[16]. The unexpected finding of elevated neurofilament levels in patient plasma during this trial are consistent with a previously unappreciated role for DLK in maintaining the integrity of the neuronal cytoskeleton[16]. The association between DLK inhibitor treatment and sensory neuropathy symptoms further suggested that DLK might directly regulate the cytoskeleton of DRG sensory neurons[16]. One major finding from our study is the direct experimental support for this latter hypothesis, as we found that acute DLK inhibition disrupts axonal neurofilament and tubulin distribution and causes an accumulation of axonal vesicles in DRG neurons (Fig. 1). These findings are being further investigated and will be reported in more detail in future studies, but one initial conclusion, consistent with[16], is that compounds that globally inhibit DLK kinase activity induce unintended side effects in neuronal axons. Nonetheless, targeting other features of DLK, including specific interactions and/or post-translational modifications, might still be of therapeutic benefit. In this study, we therefore focused on the regulation of DLK by palmitoylation, a modification that is critical for both axonal localization and signaling by DLK[8,9,17,45].

Despite providing strong support for the functional importance of DLK palmitoylation, our prior studies[8,9] had employed a DLK mutant that is never palmitoylated, and thus could not distinguish the importance of basal palmitoylation of DLK from that of any potential stimulus-dependent axonal palmitoylation. In this study we confirm that acute, TD-induced palmitoylation of DLK indeed occurs and is concomitant with DLK recruitment to axonal vesicles, consistent with a prior report[24] (Fig. 2A–D). Crucially, we further show that acute, axonal palmitoylation is critical for retrograde DLK-dependent signaling (Fig. 2E, F).

The importance of axonal palmitoylation of DLK might initially appear at odds with our prior report that the protein acyltransferase (PAT) ZDHHC17 is also critical for DLK-dependent signaling, because ZDHHC17 is Golgi-localized[9,46,47]. However, a plausible model to reconcile these findings is that ZDHHC17-dependent palmitoylation on the Golgi facilitates vesicle-based transport of DLK to distal axons, where a subset of DLK molecules is then depalmitoylated. Acute, axonal re-palmitoylation of DLK after TD, likely by a PAT distinct from ZDHHC17, is then critical for retrograde signaling. Control of distinct subcellular pools of a given palmitoyl-protein by different PATs is not without precedent[48]. These current and prior findings raised the possibility that pharmacologically either Golgi-localized or axonal palmitoylation of DLK could be neuroprotective and that blocking either or both of these processes might circumvent issues seen with cell-wide DLK inhibition.

To identify compounds that modulate DLK palmitoylation, we expanded an HCS-compatible assay that assesses the palmitoylation-dependent change in localization of transfected DLK-GFP[18]. This assay

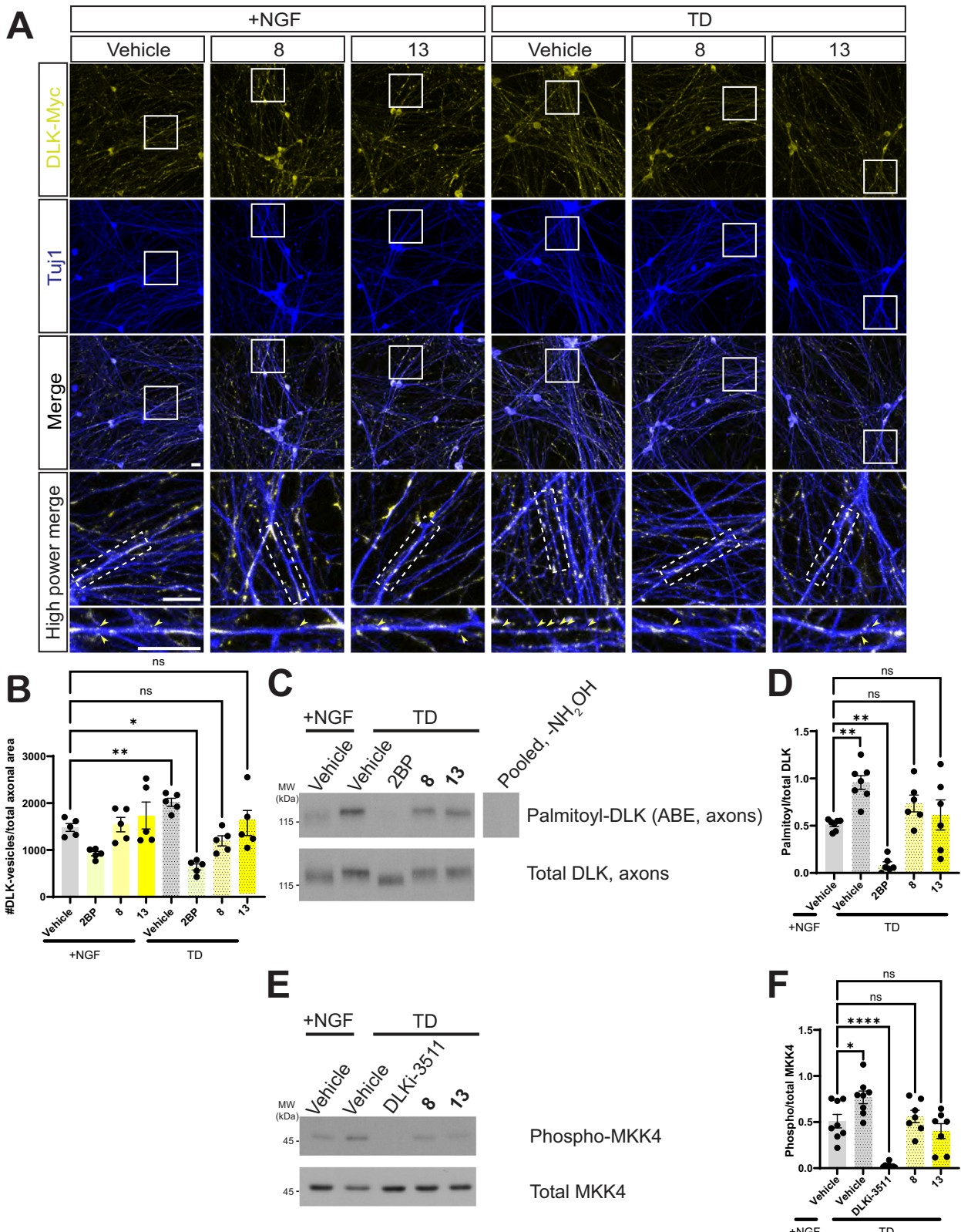

is capable of a Z-factor (a measure of assay robustness[49]) of >0.5, consistent with an excellent assay, for several readouts of DLK subcellular distribution[18]. Initial test runs of the scaled-up conditions used in the current study resulted in a slightly lower Z factor for our main Puncta/NLS readout than initially reported (0.38, Fig. S3A), but achieved a 0.50 Z factor for a parallel readout, VAI (Fig. S3B). To increase the likelihood of identifying true positive hits, we therefore

focused on compounds that were active in both readouts. When further scaled up across the entire screen, the median Z-factor was 0.4-0.5 for both readouts (Fig. S3C, D). By not excluding plates with lower Z-factor values, we recognize that our primary screen had an increased possibility of false-positives and false-negatives, but we took several complementary steps to minimize this issue. In particular, we performed extensive validation/follow-up from the primary

**Fig. 6 | Neuroprotective hit compounds prevent TD-induced DLK recruitment to axonal vesicles and palmitoylation, and activation of DLK's downstream target MKK4. A** Cultured DRG neurons were infected to express myc-tagged wild type DLK (wtDLK-myc) and left untreated or were subjected to TD in the presence of the indicated compounds or DMSO vehicle, prior to fixation and immunostaining with the indicated antibodies. The bottom two rows of images (High Power Merge) show magnified views of the boxed region in the image directly above. Arrowheads in bottom row panels indicate DLK-myc vesicle-like puncta. Scale bars, 20 μm (all panels). **B** Quantified data from A confirm that TD-induced increases in axonal DLK puncta (presumptive vesicles[24]) are prevented by **8** and **13**. Data for +NGF (vehicle), TD (vehicle) and TD (2BP) conditions are replotted from Fig. 2B. *; $p = 0.0483$; **; $p = 0.0016$, two-sided one-way ANOVA, Dunnett's post hoc test. 5 independent cultures per condition. **C** Western blots of ABE (palmitoyl-) fractions of DRG axonal lysates that had been treated as indicated prior to lysis. The right-hand lane is a side-by-side exposure from a parallel control sample omitting the key ABE reagent $NH_2OH$, run on the same gel, with intervening spacer lanes cropped.

**D** Quantified data from C confirm that **8** and **13** prevent the TD-induced increase in axonal DLK palmitoylation. Data for +NGF (vehicle), TD (vehicle) and TD (2BP) conditions are replotted from Fig. 2D. +NGF (vehicle) versus TD (vehicle): **; $p = 0.0026$; +NGF (vehicle) versus TD (2BP): **; $p = 0.0054$, two-sided one-way ANOVA, Dunnett's post hoc test. Data are plotted for the following number of independent cultures per condition: +NGF/Vehicle: 7; TD/Vehicle: 7; TD/2BP: 6; TD/**8**: 6; TD/**13**: 6. **E** Western blots, detected with the indicated antibodies, of DRG lysates that had been treated as indicated prior to lysis. **F** Quantified data from E reveal that TD significantly increases pMKK4:total MKK4 ( + NGF (Vehicle) vs. TD (Vehicle); $p = 0.0219$) while DLK inhibitor GNE-3511 (DLKi-3511) significantly reduces it (+NGF (Vehicle) vs. TD (DLKi-3511): $p < 0.0001$) but TD has no effect on pMKK4:total MKK4 in the presence of 8 or 13 (ns; non-significant). Two-sided one-way ANOVA, Dunnett's post hoc test. Data are plotted for the following number of independent cultures per condition: +NGF/Vehicle: 8; TD/Vehicle: 8; TD/2BP: 8; TD/**8**: 7; TD/**13**: 7. Data are presented as mean values +/− SEM. Source data are provided in the Source Data file.

screen in triplicate and confirmed that all hits showed dose-dependence in the primary assay prior to any testing in neurons (Table S1). In addition, it is also now appreciated that a Z-factor value of >0.5 is somewhat arbitrary, and can be challenging to achieve in cell-based assays, and that assays with Z-factor values < 0.5 can still identify useful compounds[50]. Indeed, our scaled-up assay successfully identified compounds that reduced both readouts in our primary screen, and which were then highly active in orthogonal assays in primary neurons.

Our HCS assay could potentially identify hits that predominantly affect basal DLK palmitoylation, stimulus-dependent palmitoylation, or both. Indeed, given that the majority of DLK is Golgi-localized in HEK293T cells[9,18], the screen likely favors identification of the first of these classes of compounds. It is thus intriguing that **8** and **13** were far from the most potent disruptors of DLK punctate localization in our primary screen (Table S1, Fig. S4) yet were the most effective inhibitors of palmitoyl-DLK-dependent degeneration in neuronal assays (Fig. 5). These findings are consistent with a model in which the primary mechanism of action of **8** and **13** is to selectively disrupt axonal stimulus-dependent palmitoylation of DLK, rather than basal DLK palmitoylation on the Golgi.

Mechanism of action (MOA) can be elusive to define for compounds identified in cell-based assays[51] but we have gained considerable insight into the MOA of **8** and **13**, which in turn helps explain their neuroprotective effects. In particular, both **8** and **13** have a remarkable ability to selectively prevent stimulus-dependent, but not basal, palmitoylation of the axonal pool of DLK (Fig. 6C, D, Fig. S10). Consistent with this model, both **8** and **13** also selectively block the cell biological correlate of biochemical palmitoylation, TD-induced DLK recruitment to axonal vesicles, without affecting DLK localization in healthy axons (Fig. 6A, B). Both **8** and **13** thus also act differently from 2BP, which reduces DLK palmitoylation post-TD well below the level seen in healthy (+NGF) axons (Fig. 6C, D). In contrast to its effect after TD, 2BP does not significantly reduce DLK palmitoylation in healthy axons at the time point examined (Fig. S10). These results suggest that after TD, both palmitoylation and depalmitoylation rates for DLK are increased. Importantly, and in contrast to 2BP, **8** and **13** neither decrease nor increase palmitoylation of DLK compared to basal levels, in either healthy or trophically-deprived axons. Taken together, these findings suggest that **8** and **13** inhibit stimulus-dependent DLK palmitoylation but are not broad inhibitors of cellular/axonal PATs or depalmitoylases.

We also know a considerable amount regarding what **8** and **13** do not do. In particular, neither compound blocks other key cellular events triggered by TD (shutdown or ERK and Akt signaling (Fig. S13)). Importantly, Akt inhibition is sufficient to phenocopy the effect of TD and drive DLK-dependent degeneration in DRG neurons[52] The neuroprotective action of our hit compounds is thus more likely

due to their action on a step downstream of Akt shutdown, again consistent with the notion that these compounds' key MOA involves their regulation of axonal DLK palmitoylation. In addition, neither **8** nor **13** broadly reduces palmitoylation of other axonal palmitoyl-proteins assessed (Fig. S12). Many axonal/presynaptic proteins are quite stably palmitoylated[53], a finding consistent with our results with GAP-43, $Gα_o$ and ZDHHC5, again suggesting that **8** and **13** act on dynamic enzymes/processes distinct from the bulk of axonal palmitoylation-dependent regulation. One key question that is yet to be addressed is whether TD or other axonal stimuli trigger acute palmitoylation of any other axonal proteins besides DLK, and whether any such additional palmitoylation events are important for downstream functional effects of TD, axotomy or other stimuli. Importantly, though, protection post-TD by 2BP, **8**, **13** and DLK palmitoyl-site mutation is very similar (Fig. 5[9]), suggesting that acute, axonal palmitoylation of DLK is not only functionally critical but may even be a unique palmitoylation event.

Lastly, **8** also blocks c-Jun phosphorylation triggered in response to diverse other stimuli that require DLK and its palmitoylation (axotomy, microtubule disruption by nocodazole)[8,34,35] (Fig. S14). Taken together, the properties of **8** are consistent with this compound acting as a direct inhibitor of the axonal PAT(s) for DLK. However, further testing of this model will require identification of the key relevant PAT enzyme(s). In contrast to **8**, compound **13** blocks TD- and nocodazole-induced c-Jun phosphorylation but has no effect on axotomy-induced c-Jun phosphorylation (Fig. S14). Given that palmitoylation of DLK is similarly critical for axotomy- and TD-induced DLK signaling[8,9,17], **13** is less likely to act as an inhibitor of the PAT(s) for DLK. One possibility is that **13** prevents a specific trafficking step or other cellular event, upstream of DLK palmitoylation and vesicle recruitment, that is common to TD- and nocodazole-induced signaling but which does not occur, or can be circumvented, after axotomy. This event appears to be intimately associated with acute TD-/nocodazole-induced signaling, because **13** affects neither DLK palmitoylation nor localization in healthy axons (Fig. 6A–D, Fig. S10). The distinct MOA, but similar overall protection, of **8** and **13** suggest that there are at least two potential cellular events that could be targeted therapeutically to block pro-degenerative signaling by the acutely palmitoylated axonal pool of DLK. A model consistent with the findings from this study is shown in Fig. S15. We note that in our text we have referred to GNE-3511, GDC-0134 and related compounds as DLK inhibitors because DLK is the kinase that they were developed to target. Block of retrograde signaling and cell body/axon degeneration by these compounds is highly likely to be due to on-target inhibition of DLK, because effects are phenocopied by DLK genetic knockdown or knockout[5,9,17,52]. However, we acknowledge (and schematize in Fig S15) that effects of GNE-3511, GDC-0134 and related compounds on the axonal cytoskeleton may also involve contributions from

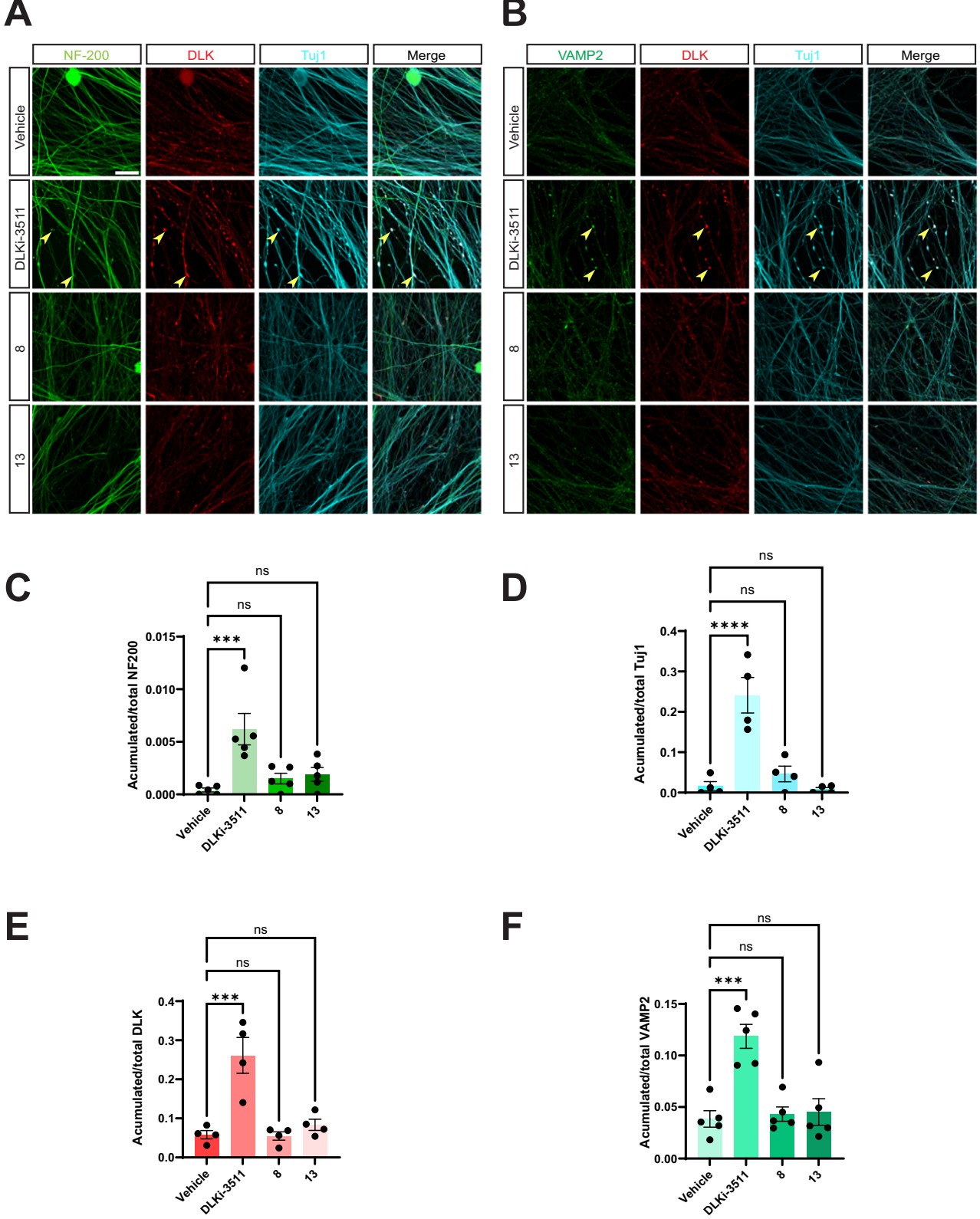

**Fig. 7 | Neuroprotective hit compounds do not phenocopy the effect of a DLK kinase domain inhibitor on axonal integrity. A** Images of cultured DRG neurons treated with DMSO vehicle, 500 nM DLKi-3511 or 10 µM of the indicated compounds for 4 hours and fixed and stained with the indicated antibodies. Yellow arrowheads highlight examples of disruption/accumulation of the respective signals. Scale bars, 20 µm (all panels). **B** As *A*, except that neurons were fixed to detect VAMP2 rather than NF-200. Quantified data from *A* and *B* confirm that DLKi-3511 causes disruptions of NF-200 (**7C** ***; $p = 0.0006$, 5 independent cultures per condition), Tuj1 (**7D** ****; $p < 0.0001$, 4 independent cultures per condition), DLK (**7E** ***; $p = 0.0003$, 4 independent cultures per condition) and VAMP2 (**7F** ***; $p = 0.0001$, 5 independent cultures per condition) in axons, but that **8** and **13** do not (n.s.; not significant). Two-sided one-way ANOVA with Dunnett's post hoc test for (**C-F**). Data are presented as mean values +/- SEM. Source data are provided in the Source Data file.

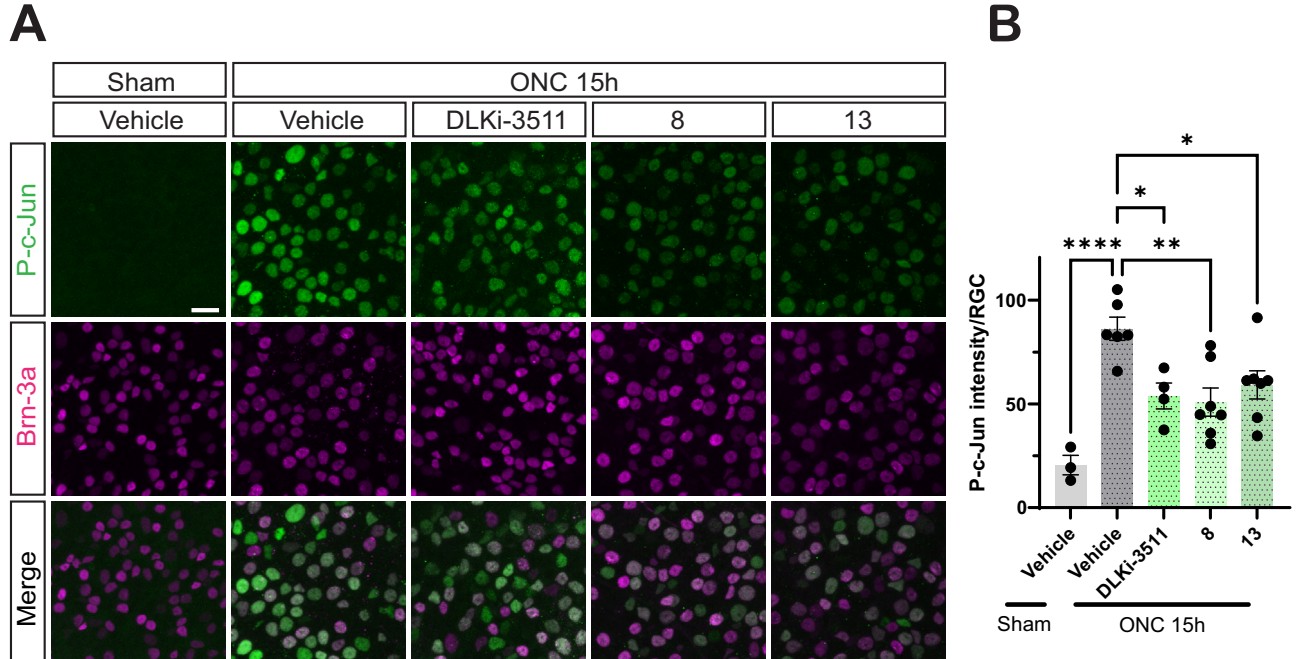

**Fig. 8 | Hit compounds reduce palmitoyl-DLK-dependent pro-degenerative retrograde signaling to a similar extent as a DLK kinase domain inhibitor in vivo. A** Images of retinas of mice that had been subjected to sham injury (1st column) or to optic nerve crush (ONC; 2nd-5th column), immediately prior to intravitreal injection with the indicated compounds or with DMSO vehicle. At 15 h post-ONC or sham injury, mice were perfused and fixed, and retinas were immunostained with the indicated antibodies. Scale bars, 20 μm (all panels). **B** Quantified data from *A*. DLKi-3511, **8** and **13** all inhibit ONC-induced c-Jun phosphorylation.

ONC (vehicle) versus ONC (DLKi-3511); *; $p = 0.0144$;), ONC (vehicle) versus ONC (**8**): **; $p = 0.0019$; ONC (vehicle) versus ONC (**13**): *; $p = 0.0181$. Sham (vehicle) and ONC (vehicle) conditions also differ significantly (****; $p < 0.0001$), two-sided one-way ANOVA, Dunnett's post hoc test. Data are plotted for the following number of mice per condition: Sham/Vehicle: 3; ONC/Vehicle: 6; ONC/DLKi-3511: 4; ONC/8: 7; ONC/13: 7. Data are presented as mean values +/- SEM. Source data are provided in the Source Data file.

DLK's close paralog leucine-zipper kinase (LZK[39,54]) and/or potential off-targets. This is an important area for future study.

DLK loss or inhibition protects both neuronal cell bodies and axons from the effects of TD[5,17,26]. Likewise, both **8** and **13** also protect cell body viability and axon integrity after extended TD, increasing the likelihood that neuroprotection by these compounds is due to their block of TD-induced DLK palmitoylation and subsequent pro-degenerative signaling. These conclusions are in part drawn from our analysis of cell body protection using a Calcein-AM assay that has been well described by others to measure DRG somal viability post-TD[55,56]. However, our quantification of the axo-protective ability of **8** and **13** used an axon integrity assay, developed in the course of this study, that relies on Interactive Watershed-based analysis to detect morphological changes in axons prior to widespread fragmentation. The readout of this assay uses phase contrast images of living neurons that can be readily acquired without expensive reagents or instrumentation and is also compatible with longitudinal time-lapse imaging. This readout could thus be particularly useful to quantify, and define mechanisms at play in, milder forms of axon degeneration, for example those seen in chronic neuropathies, which affect huge numbers of people. We suggest that our Interactive Watershed-based analysis could thus serve to complement the commonly used Axon Degeneration Index that is optimized to quantify the frank axon fragmentation induced by axotomy and other strongly axo-degenerative stimuli[27].

TD-induced retrograde signaling by (palmitoyl)-DLK in embryonic DRG neurons drives degeneration of neuronal cell bodies and axons[5,9,17] (Fig. 5B, C). The latter process depends on a transcriptional response triggered by the DLK-dependent retrograde signal, which then drives an anterograde signal from DRG cell bodies back out to axons[52]. A similar mechanism likely operates after ONC in vivo, where

degeneration of axons proximal to the crush site requires both DLK and the transcription factor cJun[10]. In both these situations, the (palmitoyl-) DLK-dependent retrograde signal is strongly pro-degenerative, for both cell bodies and proximal axons. However, DLK-dependent retrograde signaling in DRG neurons can also be pro-regenerative. For example, DLK is required for regeneration of DRG sensory axons after a sciatic nerve crush injury in adult mice, and for the accelerated rate of DRG axon growth seen after a preconditioning injury[57]. Although not directly tested, this DLK-dependent axonal growth response likely involves a specific DLK-dependent transcriptional program that is activated in DRGs after sciatic nerve injury[58]. DLK-dependent retrograde signaling in the adult peripheral nervous system is not always beneficial, though; a spared nerve injury (SNI) model also results in DLK-dependent transcription in DRG cell bodies[59], but this time the transcriptional response leads to the expression of many pain-related genes and triggers mechanical allodynia[59].

Despite these different outcomes, DLK's role in all these situations is very similar and is conceptually simple – DLK's primary action is as a stress/damage sensor that conveys signals back from damaged axons back to DRG cell bodies. Inhibiting palmitoyl-DLK-dependent retrograde signaling is thus an attractive strategy that could prevent an array of downstream consequences of axonal injury or insult. However, we recommend that a cost/benefit analysis that takes into account the disease model, developmental stage, and type of neuron/axon that must be protected should be performed when considering this approach. More broadly, we suggest that blocking key post-translational modifications or, perhaps, specific protein-protein interactions should be strongly considered alongside direct active site inhibition when seeking to therapeutically target key pro-degenerative enzymes.

When choosing a system in which to test potential efficacy of **8** and **13** as therapeutic agents we therefore focused on the response to ONC, a system in which retrograde signaling by (palmitoyl-) DLK is strongly pro-degenerative. The neuroprotective ability of **8** and **13** in cultured neurons, and their different mechanism of action compared to a DLK kinase domain inhibitor all supported this rationale. We nonetheless recognize that the block of ONC-induced p-c-Jun by **8** and **13**, while promising for potential in vivo applications, is only partial (Fig. 8A, B). There may, however, be multiple reasons for this incomplete inhibition. For example, the extent to which intravitreally injected compounds can diffuse from the RGC soma into the proximal axon, where stimulus-dependent palmitoylation is most likely to occur in vivo, is unclear and may limit their therapeutic efficacy. A second issue is that small molecules are rapidly cleared from the mouse eye after intravitreal injection[40,41] which likely limits the effective concentration of **8** and **13** in the hours after ONC. Given these issues, controlled release strategies (e.g. intravitreal implants loaded with **8** and/or **13**[60,61]) would likely be required to assess the effects of these compounds on downstream sequelae of ONC, that are also (palmitoyl-) DLK-dependent, such as breakdown of axons proximal to the crush site, and the degeneration of RGCs themselves[6,7,9,10]. If effective in preclinical models, such controlled release strategies are attractive options to treat diseases of the human visual system in which DLK is implicated[7], particularly because local intravitreal delivery and release of modulators of DLK palmitoylation-dependent signaling provides an additional way to circumvent issues with systemic DLK inhibitors[16]. An alternative, or additional, therapeutic option would be to use more expansive libraries and/or to perform combinatorial chemistry to identify more potent analogs of **8** and/or **13** or other compounds that could act as more potent in vivo neuroprotectants. Our results thus reveal a highly promising alternative approach to target specific pools of palmitoyl-DLK, which could serve as a therapeutic strategy that circumvents issues associated with global DLK inhibition.

## Methods

### Ethics
All procedures followed the National Institutes of Health guidelines, complied with ethical regulations, and were approved by the Institutional Animal Care and Use Committee (IACUC) at Temple University. For experiments involving rats and mice, food and water were provided ad libitum and animals were kept on a 12-hour light/dark cycle. Animals were kept at ambient temperatures between 21 and 23 °C and humidity between 30% and 70%.

### Molecular biology, cDNA constructs
DLK-GFP and pmCherry-NLS (the latter kindly provided by Martin Offterdinger (Addgene Plasmid #39319)) is described in ref. 18. The lentiviral vector FEW-DLK-myc is described in ref. 8. For this study the EF1 alpha promoter in this vector was replaced with a human synapsin promoter by standard subcloning, generating FSW-DLK-myc, which ensures neuron-specific DLK expression.

### Screening libraries and high content imaging
This study employed a screening library obtained by Temple University's Moulder Center for Drug Discovery. The library consists of 40,000 small molecules selected from the Maybridge Screening Collection (20,000) and Enamine Diversity Set (20,000).). Compounds were hand-selected to maximize structural diversity based on Tanimoto Index, in order to increase the chances of identifying effective pharmacophores. The library obeys Lipinski rules, with all logP values < 5 (average logP value = 3.2), <5 H-bond donors, <10 H-bond acceptors, number of rotatable bonds <8 (average # of rotatable bonds <5) and molecular weights <400 (average MW = 325). All potentially reactive molecules have been removed. Potentially reactive molecules

and PAINS candidates were not selected. Structural integrity of the library members was originally confirmed by the vendors using $^1$H-NMR and LC/MS and reconfirmed by Temple's Moulder Center using LC/MS. Compounds were originally purchased as powders and formulated into 10 mM DMSO stock solutions at the Moulder Center. 20,000 Maybridge compounds and 8,400 Enamine compounds were screened for this study. For follow-up assays in neurons, individual compounds were repurchased from Maybridge and Enamine as solids and were reconstituted as 10 mM (for cultured neuron experiments) or 100 mM (for in vivo experiments) 1000x stocks in DMSO. Details of the Maybridge and Enamine libraries are available from https://www.thermofisher.com/us/en/home/industrial/pharma-biopharma/drug-discovery-development/screening-compounds-libraries-hit-identification/high-throughput-screening-drug-discovery/properties-profile-maybridge-screening-collection.html and https://enamine.net/compound-libraries/diversity-libraries, respectively.

### Chemicals
2-Bromopalmitate (2BP) and S-Methyl methanethiosulfonate (MMTS) were from MilliporeSigma. HPDP Biotin was from Soltec Ventures. Microcystin-LR and DLK inhibitor GNE-3511 were from Cayman Chemicals. Hoechst 33342 was from Tocris Biocsciences. All other chemicals were from Thermofisher Scientific and were of the highest reagent grade.

### Antibodies
The following primary antibodies, raised in the indicated species, were purchased from Cell Signaling Technology: phospho–c-Jun (Ser63) (rabbit, #91952, used at 1:500 dilution for western blot (WB), 1:100 for immunocytochemistry and immunohistochemistry (ICC, IHC); phospho–c-Jun (Ser73) (rabbit, #3270, used at 1:500 for WB); myc (rabbit, #2278 used at 1:100 for ICC); alpha-tubulin (mouse, #3873, used at 1:2000 for WB); phospho-MKK4 (rabbit, #4514, used at 1:1000 for WB); pan-MKK4 (rabbit, #9152 used at 1:1000 for WB); Lamin A/C (mouse, #4777 used at 1:400 for WB), phospho-Akt (Thr308) rabbit #9275, used at 1:250 for WB), Akt (rabbit, #4691, used at 1:1000 for WB), pan-ERK (rabbit, #4635, used at 1:1000 for western blot), phospho-ERK1/2 (mouse #9106, used at 1:250 for WB),.. Additional antibodies were from the following indicated suppliers: anti-NGF (sheep, CedarLane, #CLMCNET-031, used at 1:40 dilution for live cell assays); DLK/MAP3K12 (rabbit, Genetex, #GTX124127, used at 1:5000 for WB); β3 tubulin (mouse, BioLegend, TUJ1, #MMS-435P, used at 1:1000 for ICC), NF-200 (mouse, MilliporeSigma #N0-142, used at 1:1000 for ICC), VAMP2 (mouse, Synaptic Systems, #104211SY, used at 1:1000 for ICC), Brn3a (mouse, Millipore Sigma, # MAB1585, used at 1:100 for IHC), GAP-43 (rabbit, Novus Biologicals NB300-143, used at 1:5000 for WB), Gα$_o$ (rabbit, ProteinTech 12-635-1-AP, used at 1:5000 for WB); ZDHHC5 (rabbit, MilliporeSigma HPA014670, used at 1:2000 for WB).

### Cell transfection
HEK293T cells were obtained from ATCC (catalog # CRL-3216) and transfected using a calcium phosphate-based method as in ref. 62.

### High Content Screening (HCS) assay
HCS assay was performed essentially as in ref. 18. Briefly, HEK293T cells were seeded in poly-lysine coated 96 well plates (Greiner Bio-One, black walled chimney-wells), transfected as above and treated with 2BP (10 μM final concentration), library compounds (10 μM final concentration) or DMSO vehicle control at 2 h post-transfection. Maybridge or Enamine library compounds (1 μL) were spotted onto 96 well plates at 10 mM in DMSO and resuspended in 200 μL pre-warmed DMEM. 40 μL of diluted compound was then added to cells in 160 μL of DMEM (containing glutamax, 10% FBS and antibiotics). Cells were returned to a tissue culture incubator for a further 14 h at 37 °C.

Medium was then aspirated and cells were fixed in 4% PFA (1x PBS) for 20 mins at RT, washed once with PBS and stained with 300 nM DAPI for 5 mins at RT, followed by 2 washes of PBS.

## High content imaging

High Content imaging was performed as in[18] using an ImageXpress micro high content imaging system (Molecular Devices, Downingtown, PA) driven by MetaXpress software. Six images per well were acquired in each of three channels (DAPI, FITC, TRITC) at 10X magnification in an unbiased fashion.

## Analysis of high content imaging data

Images were analyzed as in[18] using the MetaXpress Multiwavelength Scoring (for mCherry-NLS signals) and Transfluor modules (for DLK-GFP signals). Data were exported to a spreadsheet using the AcuityXpress software package (Molecular Devices). Three metrics were used: DLK puncta (Total Puncta Count option, from DLK-GFP signal), DLK vesicle average intensity (VAI; the intensity of the punctate DLK-GFP signal) and total number of transfected cells (from mCherry-NLS signal). The first and last of these metrics were combined to calculate DLK-GFP Puncta per NLS (P/NLS).

## Initial test of HCS assay robustness in scaled-up format

To test the robustness of our HCS assay when scaled up from our pilot screen[18], each of the first four screening runs contained an additional two 96-well plates of HEK293T cells, transfected to express wtDLK-GFP and mCh-NLS as above. Alternating blocks of 3×4 wells in each plate were treated with either DMSO vehicle or 2BP (positive control). After fixation and analysis as above, Z-factors were calculated for the P/NLS and VAI readouts for each plate.

## Analysis and inclusion/exclusion criteria for scaled-up HCS assay

Compounds that reduced P/NLS by greater than 30% of the average of vehicle-treated controls for each run were excluded from analysis due to likely cytotoxicity and/or broad effects on transcription, translation or protein stability. Compounds that increased P/NLS by >1.8 fold were also excluded from further analysis. Compounds that reduced DLK-GFP P/NLS and VAI by 2 times the standard deviation (2 SD) of the mean of all remaining determinations were considered Hits. This calculation was performed on a running basis in order to follow up initial hits while the primary screen was ongoing. The 2 SD cut-offs plotted in Fig. 3 reflect the final calculated SD values after all compounds had been screened. Follow-up assays of hit compounds were performed as above except that 10 mM stocks of compound were manually diluted to 3 mM and 1 mM in DMSO and were added to triplicate wells (10 μM, 3 μM, 1 μM final concentration) of a 96-well plate containing transfected cells.

## DRG conventional, spot and microfluidic cultures

Primary dorsal root ganglion (DRG) neurons were isolated from embryonic day 16 (E16) Sprague Dawley rat embryos of both sexes. All procedures followed the National Institutes of Health guidelines and were approved by the Institutional Animal Care and Use Committee (IACUC) at Temple University. Conventional/mass cultures were plated on either tissue culture plastic or glass coverslips pre-coated with poly-lysine and laminin, as in ref. 8. Spot cultures were plated on tissue culture plastic pre-coated with poly-lysine and laminin similar to[54]. Microfluidic cultures were prepared as in refs. 8,31, based on a design described in ref. 63.

## Trophic deprivation assay in conventional cultures

Conventional DRG cultures, prepared as above, were treated at 5 days in vitro (DIV 5) with either DMSO or 10 μM of hit compounds for 1 hour prior to trophic factor deprivation (TD). For all TD experiments, NGF-containing medium was replaced with fresh Neurobasal medium lacking NGF but containing B27 supplement plus sheep anti-NGF antibody (25 μg/ml), in the continued presence of drug or vehicle. For biochemical experiments, cells were lysed 2.5 h post-TD in SDS-PAGE loading buffer and then processed for subsequent SDS-PAGE and immunoblotting analysis. For immunocytochemistry, cultures were fixed 3 h post-TD. TD-induced responses in conventional cultures require the retrograde motor protein dynein[9] so this assay likely directly assesses retrograde axonal signaling, a process known to require palmitoyl-DLK[8,9].

## Retrograde signaling assay in microfluidic cultures

E16 DRG neurons were dissected as above and plated in microfluidic chambers as in ref. 8,17, based on a design described in ref. 63. Selective TD of distal axonal compartments was performed on DIV8-10 and cultures were processed for immunocytochemistry as in ref. 17.

## Acyl biotin exchange (ABE) palmitoylation assay from axonal fractions of DRG neurons

E16 DRG neurons were plated as spot cultures and at DIV9 were pre-treated for 1 h with either vehicle, 2BP, or hit compounds. TD was then performed in the continued presence of hit compounds or vehicle. 2 h later, a biopsy punch was used to separate the axon and soma fractions, both of which were immediately denatured in ABE lysis buffer[62]. ABE was performed essentially as in ref. 62, except that after blocking with MMTS, cultures were mixed with 1 ml of 0.08% (v/v) carrier protein (heat inactivated FBS that had itself been pre-blocked with MMTS). Protein was precipitated by adding chilled acetone (80% final [v/v]), left overnight, and MMTS was then removed by two sequential washes with 80% [v/v] acetone. Pellets were resuspended in 4% SDS buffer (4% [w/v] SDS, 50 mM Tris pH 7.5, 5 mM EDTA plus PIC) and a fraction was removed as an "Input" sample. Inputs were taken out in dilution buffer (50 mM HEPES, 1% [v/v] Triton X-100, 1 mM EDTA, 1 mM EGTA) plus PIC, 150 mM NaCl, and 5x SDS sample buffer with β-mercaptoethanol (BME) (Millipore-Sigma). Samples were split in two and incubated for 1 h rotating in the dark at room temperature in 1x Protease Inhibitor Cocktail (Boehringer), 1 mM HPDP-biotin (Soltec Ventures, Beverly, MA), 0.2% Triton X-100, plus either 1 M hydroxylamine pH 7.5, or 50 mM Tris pH 7.5. Samples were acetone-precipitated as above to remove hydroxylamine/Tris and HPDP-biotin and pellets were resuspended in ABE lysis buffer plus PIC and diluted 1:20 in dilution buffer supplemented with 150 mM NaCl, 4 μg/ml Leupeptin and 1 mM Benzamidine. Biotinylated proteins were captured by overnight incubation with high capacity neutravidin-conjugated beads (Thermo Fisher Scientific) at 4 °C. Beads were washed twice with dilution buffer containing 0.5 M NaCl, and twice with dilution buffer alone. Leupeptin and Benzamidine were added to all washes. Proteins were eluted from beads by addition of 1% (v/v) BME, 0.2% [w/v] SDS, 250 mM NaCl in dilution buffer and incubated for 10 min at 37 °C. Supernatants were removed and denatured by adding one-fifth volume of 5× SDS sample buffer. Samples were boiled and subjected to SDS-PAGE and Western blotting.

## Lentiviral preparation and infection of DRG cultures

VSV-G pseudotyped lentivirus was prepared as in ref. 62, with minor modifications. Briefly, Briefly, HEK293T cells were cotransfected with lentiviral vector plus VSV-G, pMDLg and RSV-Rev packaging plasmids using a calcium phosphate-based method. Supernatant containing virus was harvested at 48 h and 72 h post-transfection, concentrated by ultracentrifugation, resuspended in Neurobasal medium and used to infect dissociated neurons. The minimum amount of virus needed for subsequent immunocytochemical detection of virally expressed wtDLK-myc was determined in pilot studies and that amount was then infected on the second day in vitro (DIV2). On DIV5, infected cultures were pre-treated with DMSO vehicle or inhibitors for 1 h, subjected to

TD for 3 h in the continued presence of vehicle or inhibitors, and subsequently fixed and immunostained.

## Immunocytochemistry of cultured DRG neurons

Immunostaining of dissociated DRG neurons cultured on coverslips was performed essentially as described[17]. Briefly, coverslips were rinsed once with 1× recording buffer [25 mM Hepes (pH 7.4), 120 mM NaCl, 5 mM KCl, 2 mM CaCl2, 1 mM MgCl2, and 30 mM glucose] and fixed for 10 min in 4% (w/v) PFA/sucrose diluted in PBS at room temperature. Samples were permeabilized in PBS containing 0.25% (w/v) Triton X-100 for 10 min at 4 °C, blocked with PBS containing 10% (v/v) normal goat serum (SouthernBiotech, 0060-01) for 1 hour, and incubated in primary antibodies overnight at 4 °C in blocking solution. After three PBS washes, coverslips were incubated for 1 hour at room temperature with Alexa Dye–conjugated fluorescent secondary antibodies diluted in blocking solution, prior to four final PBS washes and mounting using FluorSave reagent (MilliporeSigma). Images shown in Fig. 1 and Fig. 7 were acquired using a Nikon C2 inverted confocal microscope with an oil immersion objective (60×, 1.4 NA). Acquisition parameters (laser power, gain and offset) were kept constant between all conditions. Maximum intensity projections were generated using NIS Elements software. Images in Figure 2A, Fig. 6 and Fig. 8 were acquired using a Leica SP8 confocal microscope with 40× oil immersion objective using Leica LASX software. Images in Fig. 2E were acquired using a Nikon 80i fluorescence microscope with a 10×, 0.3 NA objective using NIS Elements software. Images were exported to ImageJ/Fiji for analysis.

## Analysis of images from cultured DRG neurons

Images of endogenous DLK, VAMP2, NF200 and Tuj1 were auto-thresholded in NIH ImageJ/Fiji. The *Analyze Particles* function was then used to detect axonal accumulations of size 200-1500 pixels and with circularity 0.2-1.0 (for DLK, VAMP2) or 0.4-1.0 (for NF200, Tuj1) The percentage of axonal area occupied by these accumulations (total particle area) was then expressed as a fraction of the total axonal area as defined by each channel in the thresholded images.

For analysis of DLK-myc images, DRG cell bodies were manually identified and cleared from each image in ImageJ/Fiji, to leave only axonal signals. Images were auto-thresholded and puncta of size 4.0-10.0 pixels and circularity 0.70-1.00 were counted. The number of puncta was normalized to the fraction of the total area of the field occupied by axons, determined from auto-thresholded images from the Tuj1 channel.

## Assays of DRG neuron cell body viability and axon integrity

After performing TD at DIV5 as above, DRG cultures were returned to the incubator for 45 h. Medium was then aspirated, and neurons were incubated for 5 minutes at 37 °C with Calcein-AM UltraPure grade (CAS 148504-34-1; 1 mg/ml in DMSO) at a 1:1000 dilution in Neurobasal (NB) medium. Calcein-AM-containing medium was then aspirated, and neurons were subsequently incubated with Hoechst 33342 (10 mg/ml in dH2O stock) at a 1:2000 dilution (8 μM final concentration) in 1× PBS for 5 minutes at 37 °C. Phase contrast, and blue and green channel fluorescent images (Hoechst 33342, Calcein-AM fluorescence, respectively) were then acquired using an EVOS M5000 microscope (20× objective, 0.45NA) in an unbiased manner by imaging the center of the field and then moving to capture five random, non-overlapping fields per well. Calcein-AM images were auto-thresholded using the *Default* mode in ImageJ/Fiji. Hoechst images were auto-thresholded using the *Moments* mode. A mask of overlapping Calcein-AM and Hoechst signals was created using the Image Calculator's *AND* function to identify and count double-positive (i.e. viable) cells. The *Analyze Particles* algorithm was applied to identify these viable cells, using size parameters of 500 to infinity pixels and circularity of 0.8 - 1.0. IC$_{50}$ values for compounds in this assay were calculated from an 8-point serial dilution of compounds of interest (0.3125 μM – 20 μM, technical triplicate determinations, two biological replicates per concentration). The number of Calcein-AM/Hoechst 33342 double-positive cells per field for each concentration was then plotted using a non-linear curve fit model in GraphPad Prism to determine IC$_{50}$ values.

To quantify axonal integrity, concurrently captured phase contrast images were processed in ImageJ/Fiji. Cell bodies within the field were manually deleted and images were then inverted and processed using the Interactive H Watershed PlugIn (seed dynamics 40, intensity threshold 200, peak flooding 100%, splitting disabled) to generate a mask. The *Analyze Particles* algorithm was then applied to the watershed-processed images to identify elongated, continuous axonal structures (400-infinity pixels, circularity 0.0 - 0.3). See Fig. S6 for examples of these image processing steps.

## Transfection and Hoechst 33342 Staining in HEK293T Cells

To compare the effect of **8** and **13** with ketoconazole (the most promising hit compound from our prior pilot screen[18]), HEK293T cells were cultured on poly-lysine-coated 6-well plates and transfected to express DLK-GFP, using a calcium phosphate-based method as in ref. 18. At 4 h post-transfection, cells were recovered in medium containing either vehicle, 20 μM 2BP, 500 nM DLKi-3511, 10 μM Ketoconazole, 10 μM **8**, or 10 μM **13**. 16 h later, the culture medium was aspirated, and cells were incubated for 5 minutes with 8 μM Hoechst 33342 in PBS to stain nuclei. Wells were then washed three times with PBS before imaging on an EVOS M5000 microscope (20× objective, 0.45NA). DLK-GFP images were subjected to *Intermodes* auto-thresholding function in ImageJ/Fiji. The threshold for Hoechst 33342 images was set at a constant value of 160 (0-255 gray levels, 8-bit images). The *Analyze Particles* function was used to measure both the number of DLK-GFP puncta (5-500 pixels, circularity 0.6 – 1.0) and the number of Hoechst-positive nuclei (500-infinity pixels, circularity 0.8 - 1.0).

## Permeability assay in MDCK-MDR1 Cells

Bidirectional MDCK-MR1 assay, a predictor of permeability and CNS penetration, was performed using standard procedures[64], commercially available pre-plated cells (Pharmaron, Exton, PA) and 1 μM substrate concentrations to minimize transporter saturation. The MDCK-MR1 cell line was also used to monitor p-glycoprotein efflux liability.

## Determination of maximum aqueous solubility

Compounds were assessed for their solubility at pH 7.4 using the Millipore MultiScreenTM Solubility filter system (Millipore, Billerica, MA). Analysis was performed by LC/MS/MS on a Waters Xevo TQ instrument (Waters, Milford, MA). MS/MS analyses use positive or negative electrospray or APCI ionization. Assay acceptance criteria was 20% for all standards and 25% for the LLOQ. Results are reported as the maximum concentration of test compound obtained.

## Liver microsomal stability assay

Compounds were assessed for their stability in CD-1 mouse liver microsomes by incubating them at 37 °C in the presence or absence of an NADPH regenerating system according to standard procedures[65]. Analyses were performed by LC/MS/MS. Results are reported as half-lives (t1/2) for studies including NADPH and percent remaining after 60 minutes for studies excluding NADPH.

## Inhibition of CYP450 metabolizing enzymes

Compounds were assessed for their ability to inhibit the three major human cytochrome P450 enzymes, 3A4, 2D6 and 2C9. Expressed enzymes (Corning Gentest, Tewksbury, MA) were used to minimize non-specific binding and membrane partitioning issues[66]. The 3A4 assay used midazolam as the substrate and was analyzed using LC/MS/MS. The 2D6 and 2C9 assays used fluorescent substrates

and were analyzed on an Envision plate reader (ThermoFisher, Waltham, MA).

## Intravitreal injection and optic nerve crush

Six-week-old C57Bl/6 mice of both sexes (14 male, 13 female, assigned randomly across conditions) were obtained from Jackson Laboratories and used for this study. Mice were first were anesthetized with 0.01 mg of xylazine and 0.08 mg of ketamine per gram of body weight prior to intravitreal injection of the right eye with 2 μl of either DLKi-3511 (20 μM final concentration), **8** or **13** (100 μM) or an equivalent volume of DMSO vehicle, all diluted in 0.5× sterile PBS containing 1% (w/v) sorbitol and 5% (v/v) glycerol. The injection needle was carefully inserted behind the orthoptic lens and into the vitreous chamber to avoid damaging the lens. The left eye was left untreated. Immediately after intravitreal injection, while mice were still anesthetized, optic nerve crush (ONC) was performed as in ref. 9. In the sham group, the right eye was intravitreally injected with 2 μl of DMSO/buffer as above but was not subjected to ONC. Fifteen hours after ONC or sham injury, mice were transcardially perfused with 4% PFA in 1× PBS. Eyeballs were postfixed for an additional two hours before dissection of the retinas. Whole-mount retinal staining was performed as in ref. 9 using DAPI and antibodies raised against p-c-Jun(Ser63) and the RGC marker Brn3A. In accordance with *Nature Communications* guidelines, no post hoc sex- and gender-based analysis was performed due to low sample size when results were segregated by sex.

## Image acquisition and analysis of ONC samples

DAPI, p-c-Jun and Brn3A signals from flat-mounted retinas were acquired using a Leica SP8 confocal microscope with a 40× oil immersion objective. Maximum intensity projections were generated from confocal z-stacks and individual RGC nuclei were then identified by creating a mask of the overlapping DAPI and Brn3a signals in Fiji/ImageJ, using the Image Calculator *AND* function. This nuclear mask was then used to calculate the average intensity of the phospho–c-Jun signal in each individual RGC nucleus. For each experimental eye, phospho–c-Jun intensity within each RGC was calculated for three separate fields of view, which were then averaged to produce a single *n* per eye.

## Experimental replicates and statistical analysis

For all experiments using cultured neurons, *n* (indicated as individual data points in each Figure) reflects the indicated number of cultures, each from different dissections. In some cases, replicate determinations from a single dissection were performed side-by-side and averaged to give a single biological *n*. For immunocytochemical studies in neurons, three to four images per coverslip were acquired and analyzed to produce one biological replicate for plotting. For dose-response curves, each data point represents the average of two biological replicates, with each biological replicate consisting of technical triplicates, which were initially averaged.

Statistical analysis was performed using GraphPad Prism. Two-group comparisons of normally distributed variables were conducted using a t-test, applying Welch's correction for unequal variances or sample sizes. ANOVA was used for multi-condition experiments, with Dunnett's post hoc test for comparisons against an indicated control condition. For non-normally distributed continuous variables, the Mann-Whitney U test was used for two-group comparisons, and the Kruskal-Wallis test with Dunn's test for multiple comparisons for more than two groups.

## Reporting summary

Further information on research design is available in the Nature Portfolio Reporting Summary linked to this article.

## Data availability

The minimal dataset necessary to interpret, verify and extend the research in this study is provided in the Supplementary Information and Source Data files. Source data is provided with this paper as a Source Data file. Source data are provided with this paper.

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

## Acknowledgements

We thank Natasha Hesketh for assistance with DRG cultures and invaluable suggestions, Dr. Azita Minaei for insightful comments on the manuscript, Luiselys Hernandez for help with characterization of hit compounds in neurons and Dale Martin for initial optimization of DLK screening. We appreciate the invaluable help of Dr Marlene Jacobson regarding overall screening design, Dr Silvia Fossati for assistance with EVOS microscope imaging for neurodegeneration assays and Dr. Seo-Hee Cho (Thomas Jefferson University) for advice on retinal images. Supported by grants from NIH (R01 NS094402 and R21 EY029386, both to G.M.T.; NCI R50 CA211479 to M.B.E. and NCI Core Grant P30 CA006927 to Fox Chase Cancer Center), by Shriners' Childrens (#85190 PHI and #87400 PHI, both to G.M.T.) and by BrightFocus Foundation (G2019267, to G.M.T.).

## Author contributions

Conceptualization: G.M.T.; Experimentation/Investigation: X.Z., H.J., J.N., S.M.H., B.N.R., J.G., M.B.E., G.M.T.; Data acquisition: X.Z., H.J., J.N., S.M.H., B.N.R., J.G., M.B.E., G.M.T.; Data curation: X.Z., H.J., S.M.H., J.G., M.B.E., W.E.C., G.M.T.; Methodology: X.Z., H.J., J.N., S.M.H, B.N.R., M.B.E., G.M.T.; Writing: X.Z., H.J., G.M.T.; Funding acquisition: M.B.E., G.M.T.; Resource: M.B.E., W.E.C., G.M.T.; Supervision: W.E.C., G.M.T.

## Competing interests

A Patent Application No. 16/631,969 (National Stage Application of PCT/US18/42620) related to the screening method used in this manuscript was jointly filed by Temple University and Shriners Hospitals for Children. Authors S.M.H., J.N. (co-inventors) and G.M.T. (inventor) are named in the patent application. The patent application is being overseen by Temple University in accordance with its appropriate policies. The remaining authors declare no competing interests.
