## [Transparent Peer Review file · Nature Communications]

Inhibiting acute, axonal DLK palmitoylation is neuroprotective and avoids deleterious effects of cell-wide DLK inhibition

Corresponding Author: Dr Gareth Thomas

Version 0:

Reviewer comments:

Reviewer #1

(Remarks to the Author)

The manuscript by Zhang et al reports on the discovery of novel inhibitors blocking stress-induced palmitoylation of the MAP3K DLK. DLK signaling has a significant contribution to neurodegeneration and available inhibitors target kinase activity with notable collateral side effects that diminish therapeutic impact. This study demonstrates clear potential for targeting specific regulatory events in activation of a DLK stress signaling complex thereby bypassing deleterious impacts on the cytoskeleton.

Overall, the manuscript is timely and relevant. It provides several noteworthy discoveries: 1) describing cytoskeletal toxicity from DLK inhibitors 2) axonal DLK palmitoylation in response to trophic factor deprivation (TD) 3) identification of inhibitors blocking TD-induced DLK palmitoylation and suppression of neurodegeneration. The experimental design is largely thorough and logical. Observations are quantified with sufficient replicates. In vivo studies in Figure 8 complement cell culture studies and provide clinical relevance. Discoveries in the manuscript will benefit the field and open new lines of research.

There are additional experiments that would strengthen conclusions reported in this manuscript and boost relevance to the field of neuronal palmitoylation.

1) In ABE studies performed in Figure 6C, did the authors evaluate other proteins known to be palmitoylated? The authors previously showed that JNK2 or JNK3 are palmitoylated and comigrate with DLK in axons under steady state conditions (Niu et al Sci. Signaling 2022). Is the entire MAPK complex palmitoylated in the axon during TD or just DLK? What about palmitoylated proteins not associated with DLK signaling such as GAP43 or other examples? Do the key inhibitors selectively block DLK palmitoylation in axons? The outcome of these experiments will add substantial mechanistic insight.

2) The authors conclude their inhibitors selectively target TD-induced DLK palmitoylation. They provide numerous examples wherein these chemicals block TD-induced signaling and neurodegeneration. This conclusion would be strengthened by a more thorough analysis of DLK palmitoylation under normal (NGF+) conditions. There is quantification of DLK-myc puncta formation in Figure 2B. Evaluating DLK palmitoylation using the ABE assay under NGF+ conditions after inhibitor treatment would be a strong, complementary approach. Evaluating phospho-MKK signaling in NGF+ conditions with these inhibitors is also worthwhile, though the ABE assay would be most valuable.

3) Analysis of neurodegeneration in Figure 5 can be strengthened via consideration of the two points below.

a) Figure 5A. Phase contrast images are helpful in the extremes (e.g. Vehicle + TD versus Compound 8). However, these images and the selected high magnification examples are ambiguous. It is hard to discern differences between Compounds 6, 13, and 20 in the magnified images, though neurodegeneration scores suggest Compounds 6 and 20 are less effective. Is there a more quantitative readout that can be employed here? It is unclear if quantification in Figure 5B is accomplishing that due to the vague description of the metric (see next comment). Alternative approaches could include Ca²⁺ influx, Annexin V, propidium iodide staining.

b) Figure 5B. "Neurodegeneration Index" requires more extensive description. Are the authors referring to cell death or axon degeneration? Cited references in the Methods would imply axon degeneration. If this is the case then including example images with their corresponding score in this "Neurodegeneration Index" would improve clarity on how scoring is conducted and what constitutes a score of "4" versus a "6".

Minor Suggestions:

Figure 2. What is the timeframe of trophic factor deprivation in 2A & 2B

Figure 6A – including example images of DLK-myc localization in NGF+ conditions treated with compound 8 and/or 13 is appropriate here and reinforces a key conclusion of the study.

Reviewer #2

(Remarks to the Author)

The manuscript from Zhang et al. makes significant mechanistic and therapeutic advances in our understanding of retrograde injury signaling by the neuronal stress kinase DLK. DLK is a MAP3K that plays a key role in transmitting injury signals from the axon to the nucleus, resulting in either pro-regenerative or pro-apoptotic outcomes depending on the cell type and insult. These insults include axotomy, cytoskeletal disruption, and trophic factor deprivation (TD, which is studied here). DLK is an exciting therapeutic target for several neurodegenerative disorders. DLK inhibition was tested in a phase 1 trial, but the trial was stopped due to toxicity issues. Prior to this study it was known that palmitoylation was important for DLK signaling from the axon, but here the authors a) show that TD induces an acute change in DLK palmitoylation and vesicular localization showing this is a relevant regulatory mechanism for DLK signaling b) do a large scale phenotypic drug screen for compounds that block stimulus-induced palmitoylation c) show that these compounds block TD-induced palmitoylation of DLK as well as downstream signaling and cell death without blocking the basal activity of DLK and so avoiding the basal neurotoxicity of DLK kinase inhibitors. The authors provide compelling new mechanistic insights (injury triggers palmitoylation and vesicle association of DLK, hence likely identifying a key early step in retrograde injury signaling) and therapeutic insights (targeting stimulus-dependent palmitoylation of DLK may be as effective but much safer than targeting its kinase activity). This manuscript is a significant advance for the field.

While the manuscript is terrific, there are some straightforward experiments that could be done that would greatly enhance the impact of the work. At present, the molecular mechanism of the identified drugs is not known. While identifying the molecular target of the drugs is well beyond the scope of this publication, there are simple experiments that would greatly expand our understanding of the mechanism and specificity of these drugs. The big question that is not addressed is whether these drugs influence trophic-factor deprivation induced signaling with DLK palmitoylation as one readout, or do they impact DLK signaling more selectively with TD as just one example upstream activator?

a) There are many other molecular readouts of trophic factor deprivation (like activated NGFR retrograde transport). Do these two drugs block non-DLK, trophic factor deprivation signals?

b) In the DRG in vitro systems used in this study, other upstream insults besides TD lead to DLK-dep retrograde signaling (as assayed by pJun in the nucleus). These include axotomy and cytoskeletal disruption (usually via nocodazole). Do these two drugs block DLK-dep retrograde signaling caused by these disparate upstream signals?

The outcome of these two suggested experiments will greatly enhance our understanding of the mechanism (and potential therapeutic utility) of these drugs.

Minor point: The authors use DRGs for almost all their studies, yet their discussion and references focus primarily on the role of DLK in retrograde signaling in RGCs. A sentence or two discussing DLK in DRGs is warranted since this is the primary cell type under investigation.

Reviewer #3

(Remarks to the Author)

NCOMMS-24-35685

"Novel inhibitors of acute, axonal DLK palmitoylation are neuroprotective and avoid the deleterious side effects of cell-wide DLK inhibition"

[Key Results]

Based on the role of the dual leucine-zipper kinase (DLK) in neurodegeneration, this study explored alternative, more targeted approaches to targeting its axonal DLK palmitoylation that may avoid side effects of axonal cytoskeletal disruption associated with directly and broadly inhibiting its kinase activity. Key findings include:

1. Direct DLK kinase inhibition by GNE-3511 disrupted axonal cytoskeletal integrity in dorsal root ganglion (DRG) neurons.
2. Trophic deprivation (TD)-induced acute palmitoylation and vesicular recruitment of DLK in axons is critical for retrograde pro-degenerative signaling, as demonstrated by the effects of 2-bromopalmitate (2BP), a broad-spectrum inhibitor of cellular palmitoylation.
3. A high-content imaging screen of over 28,000 compounds in HEK293T cells transfecting with DLK identified 33 compounds that alter DLK's palmitoylation-dependent subcellular localization.
4. Two compounds (8 and 13) reducing TD-induced neurodegeneration selectively prevented TD-induced DLK palmitoylation and axonal recruitment without adverse effects on basal DLK localization or axonal cytoskeleton.

5. Intravitreal injections of compounds 8 and 13 reduced pro-degenerative retrograde signaling of c-Jun phosphorylation in a mouse model of optic nerve crush (ONC) injury.

These findings provide proof of concept for a novel and potential neuroprotective strategy that selectively targets DLK palmitoylation rather than its kinase activity.

[Validity]

The data interpretation and conclusions are generally supported by various complementary techniques. The authors use various complementary techniques to support their findings, including biochemical assays, high-content imaging, in vitro validations with primary DRG cultures, and in vivo models. However, several issues impact the validity:

1. Control groups:

- The high-content screening experiments lack important controls, e.g., 2BP, GNE-3511, and the previously identified DLK palmitoylation inhibitor ketoconazole (Martin et al., 2019, Sci Rep).
- Additional controls (e.g., 2BP) in various experiments, such as TD-induced neurodegeneration and TD-induced phosphorylation on the direct DLK substrate MKK4 and pro-degenerative retrograde signaling in the mouse model of ONC injury, would strengthen claims on neuroprotection through the specific inhibition of DLK palmitoylation.

2. Compound specificity:

For the identified compounds, a more comprehensive analysis of off-target effects on other palmitoylated proteins is needed.

3. Dose-response relationships:

More detailed dose-response data for in vitro and in vivo experiments would provide valuable information about the potency and therapeutic window of the identified neuroprotective compounds.

4. Low levels of neurodegeneration in in vitro assays:

The bright-field images in Figure 5A show unexpectedly mild levels of TD-induced neurodegeneration upon 45 hours of treatment. The mild neurodegeneration observed may limit the dynamic range for assessing neuroprotective effects. An alternative time point for optimally observing maximal neurodegeneration or a more sensitive method other than bright-field imaging alone could be helpful.

5. Mechanistic insights:

More precise mechanistic insights into how the identified compounds specifically inhibit stimulus-dependent DLK palmitoylation, which affects its recruitment to axonal vesicles and retrograde signaling, are needed.

6. Inconsistency in MKK4 phosphorylation data:

The discrepancy between immunoblot data and quantitative analysis for MKK4 phosphorylation (Figure 6E vs. 6F) needs clarification. The immunoblots in Figure 6E showed little difference among the TD group and those treated with TD + compounds 8 or 13.

7. Limited scope of in vivo analysis:

- The in vivo experiments lack an assessment of RGC survival, nerve fiber integrity, DLK localization, and functional outcomes.
- Images of sham controls are missing for the ONC model.

8. Gender of animals:

The sex of animals used in experiments was not reported to address any possible gender-related differences in the outcomes.

[Originality and significance]

This work presents a novel and potentially significant approach to targeting acute, axonal DLK palmitoylation for neuroprotection. The findings are original and build upon previous research, integrating concepts of DLK's role in neurodegeneration and its regulation by palmitoylation into a therapeutic strategy. The results are likely to be of immediate interest to researchers studying neurodegeneration, axon biology, and protein palmitoylation, as well as clinicians and pharmaceutical researchers working on neurodegenerative diseases.

[Data and methodology]

The methodology is generally sound and meets expected standards. Strengths include the use of spot cultures and microfluidic chambers for axon-specific manipulations, well-designed high-content imaging screens, and in vivo validation using ONC models. Some areas for improvement include:

1. Inclusion of appropriate controls (e.g., 2BP, GNE-3511, ketoconazole) in key experiments, as mentioned earlier.
2. While the authors have previously established the high-content imaging screening platform (Martin et al., 2019, Sci Rep), the methods used to normalize and analyze screening data, including approaches to handle plate-to-plate variability, should be provided.
3. Providing a flowchart illustrating high-content screening and analyses, with details of selection criteria, would be beneficial.

4. Highlighting control groups (e.g., 2BP, GNE-3511, ketoconazole) and the 33 identified hit compounds in the dot plots of screening results (Figure 3B-C) would be beneficial. This would demonstrate the robustness of the assay and clearly show how the hits compare to known controls and the overall compound library.
5. Additional controls for compound specificity, as mentioned earlier.
6. A clear methodology of image analysis procedures for the neurodegeneration index (Figure 5).
7. Consistent presentation of scale bars in all the images presented.
8. Reporting of animal sex and consideration of potential sex differences, as mentioned earlier.

[Analytical approach]

The analytical approach is generally appropriate, but several issues need further addressing:

1. As mentioned earlier, the inconsistency between immunoblot data and quantitative analysis for MKK4 phosphorylation (Figure 6E vs. 6F) needs to be resolved.
2. More detailed dose-response analysis for compounds 8 and 13 in vitro and in vivo would be beneficial, as mentioned earlier.
3. As mentioned earlier, additional analyses of compound specificity for DLK versus other palmitoylated proteins are needed.
4. While the authors provide qualitative descriptions of axonal blebbing/beading and cell body degeneration, quantitative analyses of these features in the in vitro neurodegeneration assays would support qualitative observations (Figure 5).

[Conclusions]

While most conclusions appear robust, some aspects require additional evidence or clarification:

1. As mentioned earlier, the discrepancy in MKK4 phosphorylation data needs to be resolved to support the proposed mechanisms of action for compounds 8 and 13.
2. The in vivo neuroprotection data are limited in scope and require expansion to support translational potential claims, as mentioned earlier fully.
3. As mentioned earlier, the lack of sex-specific analysis limits the generalizability of the findings.
4. More pronounced neurodegeneration effects in in vitro assays would strengthen conclusions about neuroprotective effects.

[Suggested improvements]

1. Including additional controls (e.g., 2BP, GNE-3511, ketoconazole) in various experiments to strengthen specificity claims.
2. Providing a flowchart illustrating high-content screening and analyses, with details of selection criteria.
3. Highlighting control groups and the identified hit compounds in high-content screening results.
4. Clearly describing the neurodegeneration index calculation method.
5. Optimizing in vitro neurodegeneration assays for more pronounced degeneration.
6. Including quantitative analyses on specific neurodegeneration features, such as axon fragmentation or neurite length, and use more sensitive methods, e.g., fluorescent stainings of specific biomarkers, to assess neurodegeneration in the in vitro assays.
7. Clarifying the inconsistency in MKK4 phosphorylation data.
8. Providing a more detailed characterization of compounds 8 and 13, including off-target effects and dose-response relationships.
9. Exploring mechanisms by which compounds 8 and 13 specifically inhibit stimulus-dependent DLK palmitoylation.
10. Reporting and analyzing the sex of animals used in all experiments.
11. Including the images of sham controls for all in vivo experiments.
12. Expanding in vivo experiments to include RGC survival, nerve fiber integrity, DLK localization, and functional assessments.

[Clarity and context]

The manuscript is generally well-written with appropriate context. Improvements could include:

1. A more detailed explanation of the high-content imaging screen methodology, especially the normalization and handling of plate-to-plate variability.
2. A flowchart of high-content screening and analyses with key decision points and criteria.
3. Clearer highlighting of neurodegenerative features of axonal blebbing/beading and cell body degeneration in representative images.
4. A schematic diagram illustrating the proposed working model of DLK regulation by palmitoylation and compound effects.

[References]

The referencing is appropriate, citing key studies on DLK function, its role in neurodegeneration, and previous attempts at therapeutic targeting. They also properly reference their prior work on DLK palmitoylation and high-content screening platforms.

[Reviewer expertise]

The manuscript's primary focus on neurodegeneration, DLK signaling, and pharmacological screening is well within the

reviewer's expertise. While the reviewer has a general understanding of pharmacokinetics, the specific methodologies and detailed analysis of these aspects might be better evaluated by experts in those fields.

Version 1:

Reviewer comments:

Reviewer #1

(Remarks to the Author)

The authors addressed my recommendations with extensive new data. I appreciate efforts to measure JNK palmitoylation and technical challenges that preclude completing this experiment. The revised manuscript provides greater mechanistic insight and will be an important contribution to the field. I have no other comments and believe this manuscript is ready for publication.

Reviewer #2

(Remarks to the Author)

The authors have done a tremendous job responding to my concerns and the concerns of the other reviewers. I thought it was a lovely paper before--not it is really rock solid!

Reviewer #3

(Remarks to the Author)

Review comment on the revised manuscript "Novel inhibitors of acute, axonal DLK palmitoylation are neuroprotective and avoid the deleterious side effects of cell-wide DLK inhibition":

The authors have done an excellent job thoroughly addressing the key concerns raised in the original review within a reasonable scope for this study. The revised manuscript represents a significant improvement that addresses the major technical and conceptual concerns raised in the original review. The additional experiments strengthen the key conclusions while better defining the limitations and future directions.

The in vivo data remains somewhat limited in scope, though the authors provide a sound explanation regarding the technical limitations of repeated intravitreal injections. Their suggestion of controlled release approaches for future studies is reasonable.

While the mechanism of action is not fully defined, the additional experiments have provided important insights into how these compounds work. The authors appropriately note that complete mechanism elucidation would require substantial additional work that is better suited for follow-up studies.

The work now provides strong evidence for a novel therapeutic strategy targeting DLK palmitoylation, with appropriate controls and quantitative measures to support the conclusions. The remaining questions about the complete mechanism and long-term in vivo effects are reasonable topics for future investigation.

*We thank the Editor and our three external reviewers for their insightful comments on the initial version of our manuscript. We were happy to note positive comments from all three reviewers. In particular, our study was described as “timely and relevant”, with “discoveries [that] will benefit the field and open new lines of research” (R1); “terrific”, with “significant mechanistic and therapeutic advances” (R2), and “novel and potentially significant”, and “of immediate interest to researchers studying neurodegeneration, axon biology, and protein palmitoylation, as well as clinicians and pharmaceutical researchers working on neurodegenerative diseases” (R3). We nonetheless recognize that there were several important points raised, including the specificity of the effects observed, and the methods used to assess neurodegeneration/neuroprotection. We have performed a large number of new experiments and have extensively revised our manuscript to address these and other concerns. The specific efforts made to address points raised in review are highlighted below in *blue italic* font. Sections of revised text that directly address additional points are copied here in Arial non-italic font with quotation marks “ ”. We feel that these changes address the vast majority of points raised in review, and fully support our model that specifically preventing acute, axonal palmitoylation of DLK could serve as a novel therapeutic strategy that circumvents issues associated with global DLK inhibition. We thank the reviewers for their consideration of this revised manuscript.*

Reviewer #1 (Remarks to the Author)

The manuscript by Zhang et al reports on the discovery of novel inhibitors blocking stress-induced palmitoylation of the MAP3K DLK. DLK signaling has a significant contribution to neurodegeneration and available inhibitors target kinase activity with notable collateral side effects that diminish therapeutic impact. This study demonstrates clear potential for targeting specific regulatory events in activation of a DLK stress signaling complex thereby bypassing deleterious impacts on the cytoskeleton.

Overall, the manuscript is timely and relevant. It provides several noteworthy discoveries: 1) describing cytoskeletal toxicity from DLK inhibitors 2) axonal DLK palmitoylation in response to trophic factor deprivation (TD) 3) identification of inhibitors blocking TD-induced DLK palmitoylation and suppression of neurodegeneration. The experimental design is largely thorough and logical. Observations are quantified with sufficient replicates. In vivo studies in Figure 8 complement cell culture studies and provide clinical relevance. Discoveries in the manuscript will benefit the field and open new lines of research.

There are additional experiments that would strengthen conclusions reported in this manuscript and boost relevance to the field of neuronal palmitoylation.

We thank the reviewer for this overall positive assessment.

1) In ABE studies performed in Figure 6C, did the authors evaluate other proteins known to be palmitoylated? The authors previously showed that JNK2 or JNK3 are palmitoylated and comigrate with DLK in axons under steady state conditions (Niu et al Sci. Signaling 2022). Is the entire MAPK complex palmitoylated in the axon during TD or just DLK? What about palmitoylated proteins not associated with DLK signaling such as GAP43 or other examples? Do the key inhibitors selectively block DLK palmitoylation in axons? The outcome of these experiments will add substantial mechanistic insight.

This is an important issue, and a similar point was also raised by Reviewer #3. In our revised manuscript we show that palmitoylation of three axonal proteins (GAP43, the alpha subunit of the G protein G_o ($G\alpha_o$) and the protein acyltransferase ZDHHC5) does not change after TD and is not affected by our novel compounds (new Fig S12). We share the reviewer's interest in the palmitoylation of JNK2 and JNK3 reported in (Niu et al Sci. Signaling 2022) and made considerable effort to address this point. However, the JNK3 antibody used in our prior study is now unavailable and other JNK3 antibodies that we tried were insufficiently sensitive to detect JNK3 in axon-only lysates, let alone in axonal ABE fractions. Likewise, our JNK2 antibody was only sensitive enough to detect total JNK2 in axon-only lysates, not palmitoyl-JNK2 in axon only ABE fractions (perhaps because only a small fraction of JNK2 is palmitoylated; Niu et al Sci. Signaling 2022). We note that (Niu et al Sci. Signaling 2022) assessed JNK2 and JNK3 palmitoylation in whole cell lysates, and that samples were pooled from entire 12-well plates. Scaling up sample collection should theoretically allow us to detect palmitoyl-JNK2 and/or -JNK3 in axon-only ABE fractions. However, we hope the reviewer appreciates that this would require considerable effort and optimization for the multiple conditions required and that s/he would agree that this this experiment would be better left to a future, separate study, preferably at a time when a sufficiently sensitive anti-JNK3 antibody is available. We emphasize that the new results that we now show do help provide the additional mechanistic insight that the reviewer was seeking i.e. we now show that our compounds do not broadly reduce palmitoylation of other axonal proteins assessed.

2) The authors conclude their inhibitors selectively target TD-induced DLK palmitoylation. They provide numerous examples wherein these chemicals block TD-induced signaling and neurodegeneration. This conclusion would be strengthened by a more thorough analysis of DLK palmitoylation under normal (NGF+) conditions. There is quantification of DLK-myc puncta formation in Figure 2B. Evaluating DLK palmitoylation using the ABE assay under NGF+ conditions after inhibitor treatment would be a strong, complementary approach. Evaluating phospho-MKK signaling in NGF+ conditions with these inhibitors is also worthwhile, though the ABE assay would be most valuable.

This is an important point. Indeed, we now show data in a new Fig S10 that our inhibitors do not reduce basal palmitoylation of axonal DLK (in the presence of NGF). We note that at the time point examined (3 h treatment, similar to the 1h pre-incubation followed by 2h TD shown in Fig 2C and 6C) DLK palmitoylation in axons is quite stable, such that it is not even significantly reduced by the broad-spectrum inhibitor 2BP, although this condition trends towards a decrease. This finding contrasts with the marked reduction in DLK palmitoylation in the TD/2BP condition, and suggests that after TD, both palmitoylation and depalmitoylation rates for DLK are increased. We briefly note this point in our revised Discussion, while emphasizing the 'take-home' message that our compounds selectively affect TD-induced DLK palmitoylation. Specifically we write:

“Interestingly, in contrast to its effect after TD, 2BP does not significantly reduce DLK palmitoylation in healthy axons at the time point examined (Fig S10). These results suggest that after TD, both palmitoylation and depalmitoylation rates for DLK are increased. Importantly though, and in contrast to 2BP, **8** and **13** neither decrease nor increase palmitoylation of DLK compared to basal levels, in either healthy or trophically-deprived axons. Taken together, these findings suggest that **8** and **13** inhibit stimulus-dependent DLK palmitoylation, but are not broad inhibitors of cellular/axonal PATs or depalmitoylases.”

*We also followed the reviewer's suggestion to assess basal MKK4 phosphorylation in healthy axons (+NGF). In a new Fig S11 we show that neither **8**, **13** nor 2BP significantly affects basal MKK4 phosphorylation, consistent with the lack of these compounds on DLK palmitoylation under the same conditions, described above.*

3) Analysis of neurodegeneration in Figure 5 can be strengthened via consideration of the two points below.

a) Figure 5A. Phase contrast images are helpful in the extremes (e.g. Vehicle + TD versus Compound 8). However, these images and the selected high magnification examples are

ambiguous. It is hard to discern differences between Compounds 6, 13, and 20 in the magnified images, though neurodegeneration scores suggest Compounds 6 and 20 are less effective. Is there a more quantitative readout that can be employed here? It is unclear if quantification in Figure 5B is accomplishing that due to the vague description of the metric (see next comment). Alternative approaches could include Ca²⁺ influx, Annexin V, propidium iodide staining.

This important point was also raised by Reviewer #3. To address this issue, we optimized a more quantitative readout of TD-induced degeneration based on Calcein-AM staining to detect live cells (similar to PMID: 28000671 and PMID: 33372032, which also assessed degeneration in trophically deprived DRG neurons). We therefore replaced our original images with 3-channel images (phase contrast, Calcein-AM, Hoechst 33342), and used an ImageJ macro to quantify Calcein-AM-Hoechst double-positive cells per field. Results of this modified assay and analysis are shown in a new Figure 5A and 5B. We also used a similar assay to assess dose-dependence of our compounds in neurons, requested by Reviewer #3. Dose-response experiments are shown in a new Fig S9.

*The Calcein-AM/Hoechst readout assesses cell body degeneration but, in part prompted by the interest of Reviewer #3, we also assessed axon integrity in the same sets of neurons. Because axons were not fully degenerated at this time point (chosen due to optimal dynamic range for the Calcein-AM/Hoechst assay) we did not use an axon degeneration index commonly used to quantify fully fragmented axons (e.g. PMID: 19403820) but instead developed an assay using the ImageJ/Fiji Interactive Watershed plug-in to identify long sections of continuous i.e. unblebbed, phenotypically intact axons. Results from this assay largely mirrored those seen with the cell body viability assay i.e. compounds **8** and **13** were the most axo-protective. These data are shown and quantified in a revised Fig 5A and 5C. We note in our Discussion that this Interactive Watershed analysis is simple to perform and could be broadly employed to study the early steps of axon degeneration and/or in models of neuropathy in which frank axon fragmentation is less pronounced.*

b) Figure 5B. "Neurodegeneration Index" requires more extensive description. Are the authors referring to cell death or axon degeneration? Cited references in the Methods would imply axon degeneration. If this is the case then including example images with their corresponding score in this "Neurodegeneration Index" would improve clarity on how scoring is conducted and what constitutes a score of "4" versus a "6".

We refer the reviewer to our answer to point (a) above, which we hope addresses this additional concern. We have also modified our Methods section to cite references that use a

similar Calcein-AM/Hoechst measurement of cell body viability and we also fully describe our new Interactive Watershed-based analysis of axon integrity.

Minor Suggestions:

Figure 2. What is the timeframe of trophic factor deprivation in 2A & 2B

We apologize for omitting this information. Images in Fig 2A and 2B were acquired 3h after TD i.e. shortly after the increase in DLK palmitoylation shown in Fig 2C, 2D. This timepoint was chosen because Tortosa et al. (PMID: 35611591) also observed vesicle recruitment of DLK 3h post-TD. Our results and those of Tortosa et al are consistent with a model in which palmitoylation and vesicle recruitment of DLK occur at similar time points, with the former likely preceding the latter. We briefly address this point in our expanded description of Figure 2 in our revised Results section. Specifically, we write:

“...we infected DRG neurons to express wild type, myc-tagged DLK (wtDLK-myc), subjected neurons to TD and then immunostained 3h later to detect myc signal. TD increased wtDLK-myc recruitment to axonal puncta (Fig 2A, B), which, based on a prior study assessing the same timepoint (3h post-TD) are likely vesicles^{8,24}.

Figure 6A – including example images of DLK-myc localization in NGF+ conditions treated with compound 8 and/or 13 is appropriate here and reinforces a key conclusion of the study.

We appreciate the reviewer raising this point and have added these images as requested.

Reviewer #2 (Remarks to the Author)

The manuscript from Zhang et al. makes significant mechanistic and therapeutic advances in our understanding of retrograde injury signaling by the neuronal stress kinase DLK. DLK is a MAP3K that plays a key role in transmitting injury signals from the axon to the nucleus, resulting in either pro-regenerative or pro-apoptotic outcomes depending on the cell type and insult. These insults include axotomy, cytoskeletal disruption, and trophic factor deprivation (TD, which is studied here). DLK is an exciting therapeutic target for several neurodegenerative disorders. DLK inhibition was tested in a phase 1 trial, but the trial was stopped due to toxicity issues. Prior to this study it was known that palmitoylation was important for DLK signaling from the axon, but here the authors a) show that TD induces an acute change in DLK palmitoylation and vesicular localization showing this is a relevant

regulatory mechanism for DLK signaling b) do a large scale phenotypic drug screen for compounds that block stimulus-induced palmitoylation c) show that these compounds block TD-induced palmitoylation of DLK as well as downstream signaling and cell death without blocking the basal activity of DLK and so avoiding the basal neurotoxicity of DLK kinase inhibitors. The authors provide compelling new mechanistic insights (injury triggers palmitoylation and vesicle association of DLK, hence likely identifying a key early step in retrograde injury signaling) and therapeutic insights (targeting stimulus-dependent palmitoylation of DLK may be as effective but much safer than targeting its kinase activity). This manuscript is a significant advance for the field.

While the manuscript is terrific, there are some straightforward experiments that could be done that would greatly enhance the impact of the work. At present, the molecular mechanism of the identified drugs is not known. While identifying the molecular target of the drugs is well beyond the scope of this publication, there are simple experiments that would greatly expand our understanding of the of mechanism and specificity of these drugs. The big question that is not addressed is whether these drugs influence trophic-factor deprivation induced signaling with DLK palmitoylation as one readout, or do they impact DLK signaling more selectively with TD as just one example upstream activator?

We thank the reviewer for the overall positive evaluation, while recognizing that assessing additional readouts would strengthen the study.

a) There are many other molecular readouts of trophic factor deprivation (like activated NGFR retrograde transport). Do these two drugs block non-DLK, trophic factor deprivation signals?

This is an important point. We now show data in a new Fig S13 that our compounds do not prevent two other molecular readouts of trophic factor deprivation (decreases in phosphorylation (i.e. inactivation) of p42/p44 ERK and Akt). This latter result also provides additional mechanistic insight because Akt inhibition phenocopies the effect of trophic factor deprivation (TD) to drive DLK-dependent degeneration in DRG neurons (PMID: 26898330). Because our novel compounds do not prevent TD-induced Akt dephosphorylation (i.e. loss/inhibition of Akt activity), the site of action of these compounds must be downstream of this event. The action of these compounds as inhibitors of TD-induced DLK palmitoylation is consistent with this model.

We also attempted to assess endogenous NGFR (p75) levels and regulation but, as with the situation with JNK3 raised by R1, we again encountered issues with antibody sensitivity. However, other studies suggest that regulation of NGFR by TD and other triggers is far better characterized in sympathetic (SCG) neurons than the DRG neurons that we employed (e.g. PMID: 29789375; PMID: 30086304; PMID: 32004684; PMID: 9472042). Moreover, it is also possible that p75NTR/NGFR regulation in DRG is insensitive to NGF deprivation (PMID: 8841920). We hope the reviewer would agree that our ERK/Akt experiments address the main point raised i.e. our compounds do not globally block TD-induced signaling events, and that examination of effects of our compounds on NGFR signaling and transport are better addressed in a separate study focused on SCG neurons.

b) In the DRG in vitro systems used in this study, other upstream insults besides TD lead to DLK-dep retrograde signaling (as assayed by pJun in the nucleus). These include axotomy and cytoskeletal disruption (usually via nocodazole). Do these two drugs block DLK-dep retrograde signaling caused by these disparate upstream signals?

*Another excellent point. We indeed addressed the effects of our compounds on axotomy- and nocodazole-induced DLK pathway activation (assessed as suggested by c-Jun phosphorylation. As shown in our new Fig S14, both **8** and **13** block nocodazole-induced c-Jun phosphorylation (perhaps even more effectively than their effects after TD). However, while **8** also blocks axotomy-induced c-Jun phosphorylation, **13** does not, suggesting a divergent mechanism of action of these two compounds. We note that cJun phosphorylation at S63 (the JNK phosphorylation site on c-Jun that we and others have routinely assessed e.g. PMID: 26719418; PMID: 21893599) was far less robust post-axotomy than post-TD, and a ladder of S63(P)-positive bands was seen in western blots of axotomized samples. We therefore also assessed the regulation of another JNK-dependent phosphorylation site on c-Jun, S73. We observed a similar block of injury-induced S73 phosphorylation by **8**, but not **13**, further supporting the conclusion that the mechanism of action of these two compounds differs. Importantly, we make a point in our revised Discussion that because **8** and **13** likely target different proteins and/or cellular events, this raises the possibility of multiple points of intervention to prevent palmitoyl-DLK-dependent degeneration.*

The outcome of these two suggested experiments will greatly enhance our understanding of the mechanism (and potential therapeutic utility) of these drugs.

Minor point: The authors use DRGs for almost all their studies, yet their discussion and

references focus primarily on the role of DLK in retrograde signaling in RGCs. A sentence or two discussing DLK in DRGs is warranted since this is the primary cell type under investigation.

We agree with the reviewer and have more fully discussed DLK-dependent retrograde signaling in DRG neurons in the revised manuscript. Indeed, we considered this topic worthy of more than the one or two sentences suggested and now write:

"TD-induced retrograde signaling by (palmitoyl)-DLK in embryonic DRG neurons drives degeneration of neuronal cell bodies and axons^{5,9,17} (Fig 5B, C). The latter process depends on a transcriptional response triggered by the DLK-dependent retrograde signal, which then drives an anterograde signal from DRG cell bodies back out to axons⁵⁰. A similar mechanism likely operates after ONC in vivo, where degeneration of axons proximal to the crush site requires both DLK and the transcription factor cJun¹⁰. In both these situations, the (palmitoyl-) DLK-dependent retrograde signal is strongly pro-degenerative, for both cell bodies and proximal axons. However, DLK-dependent retrograde signaling in DRG neurons can also be pro-regenerative. For example, DLK is required for regeneration of DRG sensory axons after a sciatic nerve crush injury in adult mice, and for the accelerated rate of DRG axon growth seen after a preconditioning injury⁵⁴. Although not directly tested, this DLK-dependent axonal growth response likely involves a specific DLK-dependent transcriptional program that is activated in DRGs after sciatic nerve injury⁵⁵. DLK-dependent retrograde signaling in the adult peripheral nervous system is not always beneficial, though; a spared nerve injury (SNI) model also results in DLK-dependent transcription in DRG cell bodies⁵⁶, but this time the transcriptional response leads to the expression of many pain-related genes and triggers mechanical allodynia⁵⁶."

Reviewer #3 (Remarks to the Author):

NCOMMS-24-35685

"Novel inhibitors of acute, axonal DLK palmitoylation are neuroprotective and avoid the deleterious side effects of cell-wide DLK inhibition"

[Key Results]

Based on the role of the dual leucine-zipper kinase (DLK) in neurodegeneration, this study explored alternative, more targeted approaches to targeting its axonal DLK palmitoylation that may avoid side effects of axonal cytoskeletal disruption associated with directly and broadly inhibiting its kinase activity. Key findings include:

1. Direct DLK kinase inhibition by GNE-3511 disrupted axonal cytoskeletal integrity in dorsal root ganglion (DRG) neurons.
2. Trophic deprivation (TD)-induced acute palmitoylation and vesicular recruitment of DLK in

axons is critical for retrograde pro-degenerative signaling, as demonstrated by the effects of 2-bromopalmitate (2BP), a broad-spectrum inhibitor of cellular palmitoylation.

3. A high-content imaging screen of over 28,000 compounds in HEK293T cells transfecting with DLK identified 33 compounds that alter DLK's palmitoylation-dependent subcellular localization.

4. Two compounds (8 and 13) reducing TD-induced neurodegeneration selectively prevented TD-induced DLK palmitoylation and axonal recruitment without adverse effects on basal DLK localization or axonal cytoskeleton.

5. Intravitreal injections of compounds 8 and 13 reduced pro-degenerative retrograde signaling of c-Jun phosphorylation in a mouse model of optic nerve crush (ONC) injury.

These findings provide proof of concept for a novel and potential neuroprotective strategy that selectively targets DLK palmitoylation rather than its kinase activity.

[Validity]

The data interpretation and conclusions are generally supported by various complementary techniques. The authors use various complementary techniques to support their findings, including biochemical assays, high-content imaging, in vitro validations with primary DRG cultures, and in vivo models. However, several issues impact the validity:

We thank the reviewer for this comprehensive assessment, while recognizing that additional issues should be addressed to test the validity of our findings.

1. Control groups:

- The high-content screening experiments lack important controls, e.g., 2BP, GNE-3511, and the previously identified DLK palmitoylation inhibitor ketoconazole (Martin et al., 2019, Sci Rep).

*We appreciate the reviewer raising this point. We included 2BP as a positive control on all screening plates and our revised Fig 3B,C now shows the average reduction in Puncta/NLS and Vesicle Average Intensity caused by this compound during all screening runs. Because the screen was already completed, we could not readily include additional compounds (GNE-3511, ketoconazole). We therefore compared effects of 2BP, **8**, **13**, GNE-3511 and ketoconazole on DLK-GFP puncta in follow-up, side-by-side experiments under very similar conditions to the initial screen. Results of these experiments, shown in a new Fig S7, confirm that **8**, **13** and*

ketoconazole reduce DLK-GFP puncta to a similar extent, but that none is as effective as 2BP. Unexpectedly, GNE-3511 reduced DLK-GFP puncta size/intensity in some cells but caused large accumulations of DLK-GFP in others. Neither of these types of DLK-GFP distribution fell within our defined cut-offs for 'DLK-GFP puncta', so quantified data in Fig S7 thus show that GNE-3511 also reduces DLK-GFP puncta number. While there are clearly additional mechanisms at play in GNE-3511-treated HEK293T cells, we hope the reviewer would agree that defining this effect is better suited to a follow-up study.

- Additional controls (e.g., 2BP) in various experiments, such as TD-induced neurodegeneration and TD-induced phosphorylation on the direct DLK substrate MKK4 and pro-degenerative retrograde signaling in the mouse model of ONC injury, would strengthen claims on neuroprotection through the specific inhibition of DLK palmitoylation.

This is an important point. We now cite our prior work (Niu et al., Sci Signaling, 2022; PMID: 35349303) in which we showed that 2BP indeed blocks TD-induced phosphorylation of MKK4.

Specifically, we write:

*“In a prior study, 2BP also blocked TD-induced increases in pMKK4, consistent with a model in which this event requires DLK and its palmitoylation¹⁷. In contrast, but consistent with their lack of effect on basal DLK palmitoylation and localization, neither **8**, **13** or 2BP reduced basal pMKK4 levels in healthy axons (Fig S11).”*

We also now show that 2BP reduces TD-induced degeneration of DRG somas and axons (using our two updated assays, new Fig 5A-C). This latter result was surprising, given the broad array of proteins that 2BP targets, but further supports the conclusion that palmitoylation is critical for TD-induced degeneration and even suggests that DLK is the key and perhaps only, axonal protein whose depalmitoylation in the presence of 2BP underlies neuroprotection by this compound.

As suggested by the reviewer, we also assessed the effect of 2BP in the retina in the mouse model of ONC. However, in pilot experiments we found that 2BP clearly upregulated cJun phosphorylation in non-RGC cells (perhaps astrocytes and/or displaced amacrine cells) in sham-operated controls. We show this experiment for the reviewer below (“Figure A”). 2BP did appear to blunt c-Jun phosphorylation in RGCs themselves after ONC, but phospho-cJun positive non-RGC cells were also visible in this condition. This additional effect of 2BP provides further rationale to evaluate and develop compounds that more selectively target palmitoyl-DLK-dependent signaling. However, given these additional complications, we hope the reviewer understands our decision not to pursue in vivo experiments with 2BP further within

the current study. We have also not included the 2BP experiment shown below in the revised submission but, if the reviewer and/or editor feel that it is important information and should be included, we would be happy to revise the manuscript accordingly.

Figure A: Unexpected elevation of cJun phosphorylation in non-RGCs in sham-injured retinas. Images of flat-mounted retinas fixed and stained with the indicated antibodies 15h after sham injury or optic nerve crush (ONC). 2BP or vehicle control was intravitreally injected immediately after ONC or sham injury. In the sham injured condition, 2BP triggers cJun phosphorylation (P-c-Jun), mainly in cells that do not detectably express the RGC marker Brn3a (2nd column, yellow arrowheads). P-c-Jun-positive non-RGCs are also seen post-ONC (4th column, yellow arrowheads). Representative images from a total of 3 mice (1 sham injury, 2 ONC) assessed. Scale bar: 20 μ m (all panels)

2. Compound specificity:

For the identified compounds, a more comprehensive analysis of off-target effects on other palmitoylated proteins is needed.

*This is an important issue, and Reviewer #1 also asked about effect of our compounds on other palmitoyl-proteins. We now show, in a new Fig S12, that our novel compounds do not affect axonal palmitoylation of the palmitoyl-proteins GAP-43, G protein $G\alpha_o$, and the protein acyltransferase ZDHHC5. We also now show that our compounds do not reduce basal axonal palmitoylation of DLK (new Fig S10). These findings suggest that **8** and **13** do not broadly affect palmitoylation of axonal proteins.*

3. Dose-response relationships:

More detailed dose-response data for in vitro and in vivo experiments would provide valuable information about the potency and therapeutic window of the identified _ neuroprotective compounds.

*We agree with this point and now provide dose-response data, including IC50 values calculated from 8-point dilution curves, for both **8** and **13** in our assay of TD-induced cell body viability (new Fig S9A-D). We hope the reviewer would agree that a similar dose-*

response curve in vivo (which would require over 50 additional surgeries) is better suited to a follow-up study. Indeed, in informal discussions with the Editor it was agreed that these experiments are beyond the scope of the current manuscript.

4. Low levels of neurodegeneration in in vitro assays:

The bright-field images in Figure 5A show unexpectedly mild levels of TD-induced neurodegeneration upon 45 hours of treatment. The mild neurodegeneration observed may limit the dynamic range for assessing neuroprotective effects. An alternative time point for optimally observing maximal neurodegeneration or a more sensitive method other than bright-field imaging alone could be helpful.

*We appreciate this point. To address this issue, we optimized a more quantitative and more sensitive readout of TD-induced degeneration. This method uses Calcein-AM staining to detect live cells (similar to PMID: 28000671 and PMID: 33372032, which also assessed degeneration in trophically deprived DRG neurons). In an updated Fig 5A we now show 3-channel images (phase contrast, Calcein-AM, Hoechst 33342) plus merged images of Calcein-AM, Hoechst 33342 double-positive cells. In Fig 5B we show results of an ImageJ macro used to quantify double-positive cells per field. Importantly, the results with these new, more quantitative measurement, broadly mirror those seen in our original, manual quantification, with **8** and **13** providing markedly protecting DRG cell bodies from the effects of TD. We also used this optimized Calcein-AM/Hoechst assay for the dose-response experiments described above (new Fig S9A-D).*

5. Mechanistic insights:

More precise mechanistic insights into how the identified compounds specifically inhibit stimulus-dependent DLK palmitoylation, which affects its recruitment to axonal vesicles and retrograde signaling, are needed.

*This is an important point. While the precise mechanism of action (which is often challenging to define for compounds identified in cell-based screens) is still unknown for **8** and **13**, additional experiments performed in response to review have provided several key insights. For example, we now show that neither **8** nor **13** prevents TD-induced shutdown of the pro-survival ERK and Akt pathways (new Fig S13). One key conclusion from these results is that neither **8** nor **13** broadly blocks all effects of TD. Moreover, because Akt inhibition/dephosphorylation can phenocopy the effect of TD and drive DLK-dependent degeneration in DRG neurons (PMID: 26898330), the neuroprotective action of **8** and **13** is thus more likely due to their action on a step downstream of Akt shutdown. This model is*

again consistent with the notion that these compounds' key MOA involves their regulation of axonal DLK palmitoylation. As discussed above (Point #2) we also show that neither **8** nor **13** reduces palmitoylation of multiple other axonal palmitoyl-proteins (new Fig S12) i.e. these compounds do not broadly block axonal palmitoylation and/or stimulate depalmitoylation. We further show that neither **8** nor **13** reduces basal DLK palmitoylation, or pMKK4 levels in the continued presence of NGF (new Figs S10, S11). Finally, we now show that **8** prevents DLK-dependent signaling in response to multiple additional stimuli (nocodazole, axotomy) but that **13** selectively blocks nocodazole-induced cJun phosphorylation (new Fig S14). These findings are consistent with a model in which **8** broadly blocks stimulus-dependent DLK palmitoylation and signaling, perhaps by inhibiting the protein acyltransferase for DLK. In contrast, **13**, which more selectively prevents certain forms of palmitoyl-DLK signaling, may prevent a specific trafficking step or other cellular event, upstream of DLK palmitoylation and vesicle recruitment. This event would be common to TD- and nocodazole-induced signaling but would not occur, or can be circumvented, after axotomy. Importantly, the likely different points of action for **8** and **13** suggest that there are at least two proteins/processes that could be targeted therapeutically to block palmitoyl-DLK-dependent signaling. We have added a paragraph summarizing the above points to our revised Discussion. Specifically, we write:

“...**8** blocks c-Jun phosphorylation triggered in response to diverse other stimuli (axotomy, nocodazole) that require DLK and its palmitoylation^{8,32,33} (Fig S14). Taken together, the properties of **8** are consistent with this compound acting as a direct inhibitor of the axonal PAT(s) for DLK. However, further testing of this model will require identification of the key relevant PAT enzyme(s). In contrast to **8**, compound **13** blocks TD- and nocodazole-induced c-Jun phosphorylation but has no effect on axotomy-induced c-Jun phosphorylation (Fig S14). Given that palmitoylation of DLK is similarly critical for axotomy- and TD-induced DLK signaling^{8,9,17}, **13** is thus less likely to act as an inhibitor of the PAT(s) for DLK. One possibility is that **13** prevents a specific trafficking step or other cellular event, upstream of DLK palmitoylation and vesicle recruitment, that is common to TD- and nocodazole-induced signaling but which does not occur, or can be circumvented, after axotomy. Importantly, this event would appear intimately associated with acute TD-/nocodazole-induced signaling, because **13** affects neither DLK palmitoylation nor localization in healthy axons (Fig 6A-D, Fig S10). The distinct MOA, but similar overall protection, by **8** and **13** suggest that there are at least two potential cellular events that could be targeted therapeutically to block pro-degenerative signaling by the acutely palmitoylated axonal pool of DLK.”

We hope the reviewer would agree that these new findings add substantial mechanistic insight but that a more precise definition of mechanism of action is better left to a follow-up study.

6. Inconsistency in MKK4 phosphorylation data:

The discrepancy between immunoblot data and quantitative analysis for MKK4

phosphorylation (Figure 6E vs. 6F) needs clarification. The immunoblots in Figure 6E showed little difference among the TD group and those treated with TD + compounds 8 or 13.

We have replaced the original phospho- and total MKK4 blots in Fig 6E with panels that more accurately reflect the pooled, quantified data.

7. Limited scope of in vivo analysis:

- The in vivo experiments lack an assessment of RGC survival, nerve fiber integrity, DLK localization, and functional outcomes.

We appreciate and share the reviewer's interest in this point. However, the rapid clearance of small molecules from the retina (PMID: 37657528; PMID: 31430156) means that assessing these readouts, which occur several days-to-weeks post-ONC (PMID: 33207199; PMID: 23431164; PMID: 23431148; PMID: 33207199; PMID: 23431164; PMID: 23431148) would require repeated intravitreal injections of our compounds (perhaps once to twice daily for several days). We hope the reviewer would agree that the high potential for false-negative results and/or complications of interpretation due to technical issues makes these experiments best suited to a future study. However, we do discuss this issue, and suggest that controlled release strategies using intravitreal implants, could be a possible therapeutic approach, albeit one that is beyond the scope of the current manuscript. Specifically, we write:

*"...Controlled release strategies (e.g. intravitreal implants loaded with **8** and/or **13**^{57,58}) would likely be required to assess the effects of these compounds on downstream sequelae of ONC, that are also (palmitoyl-) DLK-dependent, such as breakdown of axons proximal to the crush site, and the degeneration of RGCs themselves^{6,7,9,10}. However, if effective in preclinical models, such strategies are attractive options to treat diseases of the human visual system in which DLK is implicated⁷, particularly because local intravitreal delivery and release of modulators of DLK palmitoylation-dependent signaling could circumvent issues with systemic DLK inhibitors¹⁶"*

- Images of sham controls are missing for the ONC model.

We have added these images as requested.

8. Gender of animals:

The sex of animals used in experiments was not reported to address any possible gender-related differences in the outcomes.

We have added this information as requested. DRG neurons for primary cultures were

obtained from rat embryos of both sexes, and mice of both sexes were used for in vivo studies. Regarding in vivo experiments, we now write:

“Six-week-old mice of both sexes (14 male, 13 female, assigned randomly across conditions) were used for this study...

...In accordance with Nature Communications guidelines, no post hoc sex- and gender-based analysis was performed due to low sample size when results were segregated by sex.”

Despite the Nature Communications guidelines summarized above, we nonetheless appreciate the reviewer’s interest in potential sex-based differences and so provide here a modified version of Fig 8B, with data points for males (triangles) and females (circles) indicated (“Figure B”). There is no clear difference in results based on animal sex.

Figure B: No clear difference in retinal response to optic nerve crush, or response to inhibitors, based on sex. Data from Fig 8B were plotted to indicate the sex of each mouse used (males: triangles, females: circles). There is no apparent difference in the response to optic nerve crush or the effect of inhibitory compounds, based on sex, although no formal post hoc analysis of sex-segregated data was performed, as per Nature Communications guidelines.

[Originality and significance]

This work presents a novel and potentially significant approach to targeting acute, axonal DLK palmitoylation for neuroprotection. The findings are original and build upon previous research, integrating concepts of DLK's role in neurodegeneration and its regulation by palmitoylation into a therapeutic strategy. The results are likely to be of immediate interest

to researchers studying neurodegeneration, axon biology, and protein palmitoylation, as well as clinicians and pharmaceutical researchers working on neurodegenerative diseases.

[Data and methodology]

The methodology is generally sound and meets expected standards. Strengths include the use of spot cultures and microfluidic chambers for axon-specific manipulations, well-designed high-content imaging screens, and in vivo validation using ONC models. Some areas for improvement include:

1. Inclusion of appropriate controls (e.g., 2BP, GNE-3511, ketoconazole) in key experiments, as mentioned earlier.

*This is an important point. As described above in response to the original comment, we included 2BP as a positive control on all screening plates and our revised Fig 3B,C now shows the average reduction in Puncta/NLS and Vesicle Average Intensity caused by this compound during screening runs. Because the screen was already completed, we could not readily include additional compounds (GNE-3511, ketoconazole). We did, however, compare effects of 2BP, **8**, **13**, GNE-3511 and ketoconazole on DLK-GFP puncta in follow-up side-by-side experiments under very similar conditions to the initial screen. These experiments are shown and quantified in a new Fig S7. **8** and **13** significantly reduce DLK-GFP puncta in this follow-up assay, even though (consistent with results from the primary screen) both are less effective than 2BP.*

2. While the authors have previously established the high-content imaging screening platform (Martin et al., 2019, Sci Rep), the methods used to normalize and analyze screening data, including approaches to handle plate-to-plate variability, should be provided.

We appreciate this point. Our new Figure S3A, B show the Z-factor values (a measure of assay robustness, with Z-factor >0.5 classically considered an excellent assay; PMID: 10838414) for both readouts (puncta per NLS and Vesicle Average Intensity) for initial test runs using the scaled-up conditions for the current screen (compared with Martin et al., 2019). Although our assay platform can achieve a Z-factor ≥ 0.5 for these readouts, we did not achieve this value for all plates in the main screen (plotted in a new Fig S3C, D). We nonetheless consider that the median Z-factor values across the entire screen (P/NLS: 0.41; VAI: 0.49) and our consideration of both readouts when selecting compounds for follow-up is more than acceptable for a cell-based screen, particularly when it is also now appreciated that a Z-factor value of >0.5 is somewhat arbitrary, and can be challenging to achieve in cell-based assays,

and that assays with Z-factor values <0.5 can still identify useful compounds PMID: 32749188. We summarize these points in our revised Discussion as follows:

"To identify compounds that modulate DLK palmitoylation, we expanded an HCS-compatible assay that assesses the palmitoylation-dependent change in localization of transfected DLK-GFP¹⁸. This assay is capable of a Z-factor (a measure of assay robustness⁴⁹) of >0.5, consistent with an 'excellent' assay, for several readouts of DLK subcellular distribution¹⁸. Initial test runs of the scaled-up conditions used in the current study resulted in a slightly lower Z factor for our main Puncta/NLS readout than initially reported (0.38, Fig S3A), but achieved a 0.50 Z factor for a parallel readout, VAI (Fig S3B). To increase the likelihood of identifying true positive hits, we therefore focused on compounds that were active in both readouts. When further scaled up across the entire screen, the median Z-factor was 0.4-0.5 for both readouts (Fig S3C, D). By not excluding plates with lower Z-factor values, we recognize that our primary screen had an increased possibility of false-positives and false-negatives, but we took several complementary steps to minimize this issue. In particular, we performed extensive validation/follow-up from the primary screen in triplicate and confirmed that all hits showed dose-dependence in the primary assay prior to any testing in neurons (Table S1). In addition, it is also now appreciated that a Z-factor value of >0.5 is somewhat arbitrary, and can be challenging to achieve in cell-based assays, and that assays with Z-factor values <0.5 can still identify useful compounds⁵⁰. Indeed, our scaled-up assay successfully identified compounds that reduced both readouts in our primary screen, and which were then highly active in orthogonal assays in primary neurons."

3. Providing a flowchart illustrating high-content screening and analyses, with details of selection criteria, would be beneficial.

We have now added an additional flowchart with details of selection criteria for specific steps as requested (Fig S8). We also expanded Table S1 to show the results of assessments of dose-dependence, and we completed Table S2, a Nature Communications template document, in which we outline the key features of our HCS assay.

4. Highlighting control groups (e.g., 2BP, GNE-3511, ketoconazole) and the 33 identified hit compounds in the dot plots of screening results (Figure 3B-C) would be beneficial. This would demonstrate the robustness of the assay and clearly show how the hits compare to known controls and the overall compound library.

*We have taken different approaches to address this point. First, we have now added a line on Fig 3B, 3C to show the average effect of 2BP, which was included in all screening plates. Second, we have also added a new Fig S4A,B highlighting the positions of the 33 hit compounds. This Figure shows a series of 4500 compound bins, to avoid issues of 'dot density' obscuring individual hits on the main Figure 3. Finally, for 2BP, GNE-3511, ketoconazole, **8***

and **13** we now show representative images of effects on DLK-GFP puncta in follow-up assays (Fig S7).

5. Additional controls for compound specificity, as mentioned earlier.

*We have again taken different approaches to address this point. We now show that our compounds do not affect palmitoylation of three additional palmitoyl-proteins in axons following TD (new Fig S12). We also show that our compounds do not affect basal palmitoylation of DLK (new Fig S10) or basal pMKK4 (new Fig S11). Furthermore, we show that our compounds do not prevent other signaling events triggered by NGF withdrawal (shutdown of ERK and Akt pathways) (new Fig S13). Finally, we show that **8** also blocks (palmitoyl-) DLK-dependent signaling in response to nocodazole and axotomy, whereas **13** blocks nocodazole-, but not axotomy-induced DLK signaling (new Fig S14). We hope the reviewer would agree that these new experiments provide important insights into what **8** and **13** do and do not do.*

6. A clear methodology of image analysis procedures for the neurodegeneration index (Figure 5).

We now provide a more detailed methodology for our new, more quantitative analysis of cell body viability (Methods and new Fig 5). Specifically, we write:

“Assays of DRG neuron cell body viability and axon integrity

After performing TD at DIV5 as above, DRG cultures were returned to the incubator for 45h. Medium was then aspirated, and neurons were incubated for 5 minutes at 37°C with Calcein-AM UltraPure grade (CAS 148504-34-1; 1 mg/ml in DMSO) at a 1:1000 dilution in Neurobasal (NB) medium. Calcein-AM-containing medium was then aspirated, and neurons were subsequently incubated with Hoechst 33342 (10 mg/ml in dH₂O stock) at a 1:2000 dilution (8 μM final concentration) in 1× PBS for 5 minutes at 37°C. Phase contrast, and blue and green channel fluorescent images (Hoechst 33342, Calcein-AM fluorescence, respectively) were then acquired using an EVOS M5000 microscope (20× objective, 0.45NA) in an unbiased manner by imaging the center of the field and then moving to capture five random, non-overlapping fields per well. Calcein-AM images were auto-thresholded using the “Default” mode in ImageJ/Fiji. Hoechst images were auto-thresholded using the “Moments” mode. A mask of overlapping Calcein-AM and Hoechst signals was created using the Image Calculator’s “AND” function to identify and count double-positive (i.e. viable) cells. The “Analyze Particles” algorithm was applied to identify these viable cells, using size parameters of 500 to infinity pixels and circularity of 0.8 - 1.0.”

7. Consistent presentation of scale bars in all the images presented.

We now show scale bars of a more consistent size (20 μm), as suggested, while noting that the magnification and/or format of some images differs from Figure to Figure. We are happy to make additional specific changes if requested.

8. Reporting of animal sex and consideration of potential sex differences, as mentioned earlier.

We now include information regarding animal sex and consideration of potential sex differences. Specifically, we write...

“Six-week-old mice of both sexes (14 male, 13 female, assigned randomly across conditions) were used for this study...

...In accordance with Nature Communications guidelines, no post hoc sex- and gender-based analysis was performed due to low sample size when results were segregated by sex.”

We nonetheless note that the sample sizes used for our study (n=3-7 mice per condition) are similar to, and in some cases greater than, those used in similar studies of DLK-dependent responses to ONC e.g. PMID: 23431164; PMID: 23431148; PMID: 24878510.

[Analytical approach]

The analytical approach is generally appropriate, but several issues need further addressing:

1. As mentioned earlier, the inconsistency between immunoblot data and quantitative analysis for MKK4 phosphorylation (Figure 6E vs. 6F) needs to be resolved.

We have replaced the original phospho- and total MKK4 blots in Fig 6E with panels that more accurately reflect the pooled, quantified data.

2. More detailed dose-response analysis for compounds 8 and 13 in vitro and in vivo would be beneficial, as mentioned earlier.

*We agree with this point and now provide 8-point dose-response data, including IC50 values, for both **8** and **13** in our assay of TD-induced cell body degeneration (new Fig S9). We hope the reviewer would agree that a similar dose-response curve in vivo (which would require over 50 additional surgeries) is better suited to a follow-up study. Indeed, in informal discussions with the Editor it was agreed that these experiments are beyond the scope of the current manuscript.*

3. As mentioned earlier, additional analyses of compound specificity for DLK versus other palmitoylated proteins are needed.

We now show that our compounds do not affect palmitoylation of three additional palmitoyl-proteins in axons following TD (Fig S12) and do not affect basal palmitoylation of DLK (Fig S10).

4. While the authors provide qualitative descriptions of axonal blebbing/beading and cell body degeneration, quantitative analyses of these features in the in vitro neurodegeneration assays would support qualitative observations (Figure 5).

*We agree with this point. Because axons were not fully degenerated at the time point that gave optimal dynamic range in our assay of cell body viability, we therefore developed an assay using the ImageJ/Fiji Interactive Watershed function to identify long sections of continuous i.e. unblebbed, phenotypically intact axons. Results from this assay largely mirrored those seen with the cell body viability assay i.e. compounds **8** and **13** were the most axo-protective. These data are shown and quantified in a revised Fig 5A and 5C, respectively. We note in our Discussion that this Interactive Watershed analysis could be broadly employed to study the early steps of axon degeneration and/or in models of neuropathy in which frank axon fragmentation is less pronounced. We thank the reviewer for suggesting that we address this issue more directly.*

[Conclusions]

While most conclusions appear robust, some aspects require additional evidence or clarification:

1. As mentioned earlier, the discrepancy in MKK4 phosphorylation data needs to be resolved to support the proposed mechanisms of action for compounds 8 and 13.

We have replaced the original phospho- and total MKK4 blots in Fig 6E with panels that more accurately reflect the pooled, quantified data.

2. The in vivo neuroprotection data are limited in scope and require expansion to support translational potential claims, as mentioned earlier fully.

We appreciate this point and, as requested and discussed above, began by assessing the effect of 2BP on ONC responses. However, as discussed above, we found that 2BP increased c-Jun phosphorylation in non-RGC cells in the retina, even after sham injury. While we recognize the potential importance of other readouts suggested (nerve fiber integrity, functional measurements), as discussed above, these processes are only impacted several days-to-weeks post-ONC (e.g. PMID: 33207199; PMID: 23431164; PMID: 23431148). However, the rapid

clearance of small molecules from the retina (PMID: 37657528; PMID: 31430156) means that assessing these readouts would require repeated intravitreal injections of our compounds (perhaps once to twice daily for several days) or an alternate approach (e.g. controlled release from an intravitreal implant). We hope the reviewer would agree that the high potential for false-negative results and/or complications of interpretation due to technical issues makes these experiments best suited to a future study, but we do discuss the use of intravitreal controlled release approaches to assess the effects of our compounds on downstream sequelae of optic nerve crush. Specifically, we write:

“...small molecules are rapidly cleared from the mouse eye after intravitreal injection^{38,39} which likely limits the effective concentration of **8** and **13** in the hours after ONC. Given these issues, controlled release strategies (e.g. intravitreal implants loaded with **8** and/or **13**^{57,58}) would likely be required to assess the effects of these compounds on downstream sequelae of ONC, that are also (palmitoyl-) DLK-dependent, such as breakdown of axons proximal to the crush site, and the degeneration of RGCs themselves^{6,7,9,10}. If effective in preclinical models, such strategies are attractive options to treat diseases of the human visual system in which DLK is implicated⁷, particularly because local intravitreal delivery and release of modulators of DLK palmitoylation-dependent signaling could circumvent issues with systemic DLK inhibitors¹⁶.”

3. As mentioned earlier, the lack of sex-specific analysis limits the generalizability of the findings.

We now provide more information regarding the sex of animals used, and the consideration of potential sex differences. Specifically, we write:

“Six-week-old mice of both sexes (14 male, 13 female, assigned randomly across conditions) were used for this study...”

...In accordance with Nature Communications guidelines, no post hoc sex- and gender-based analysis was performed due to low sample size when results were segregated by sex.”

However, because Nature Communications discourages conducting sex-based post hoc analysis of data if sample size is low, we did not segregate results by sex. We nonetheless note that the sample sizes used for our study (n=3-7 mice per condition) are similar to, and in some cases greater than, those used in similar studies of DLK-dependent responses to ONC e.g. PMID: 23431164; PMID: 23431148; PMID: 24878510.

Despite the Nature Communications guidelines summarized above, we nonetheless appreciate the reviewer's interest in potential sex-based differences and so provide here a modified version of Fig 8B, with data points for males (triangles) and females (circles) indicated ("Figure B"). There is no clear difference in results based on animal sex.

Figure B: No clear difference in retinal response to optic nerve crush, or response to inhibitors, based on sex. Data from Fig 8B were re-plotted to indicate the sex of each mouse used (males: triangles, females: circles). There is no apparent difference in the response to optic nerve crush or the effect of inhibitory compounds, based on sex, although no formal post hoc analysis of sex-segregated data was performed, as per Nature Communications guidelines.

4. More pronounced neurodegeneration effects in in vitro assays would strengthen conclusions about neuroprotective effects.

Our new Calcein-AM assay is well described to assess DRG neuron viability after trophic factor deprivation (e.g. PMID: 28000671; PMID: 33372032) and shows a 5-fold effect of NGF withdrawal at the timepoint assessed, which we feel is a more-than-sufficient dynamic range. The almost complete (>80%; Fig 5A, B) protective effect of compounds of interest (DLKi-3511, 2BP, 8 and 13) in this assay is more than sufficient to classify these compounds as neuroprotectants.

We recognize that degeneration of axons, assessed in phase contrast images, is less pronounced at the same time point. However, our new Interactive Watershed-based analysis

of axon integrity has unexpected advantages in that it readily detects small differences in the number of long, physically intact axons under conditions that are not suited to analysis with the widely used Axon Degeneration Index (optimized to detect fully fragmented axons). We suggest that other researchers may find that these two assays complement one another. In particular, our Axon Integrity Assay may be especially well suited to quantify, and define mechanisms at play in, milder forms of axon degeneration, for example those seen in chronic neuropathies, which affect huge numbers of people

[Suggested improvements]

1. Including additional controls (e.g., 2BP, GNE-3511, ketoconazole) in various experiments to strengthen specificity claims.

*This is an important point. We included 2BP as a positive control on all screening plates and our revised Fig 3B,C now shows the average reduction in Puncta/NLS and Vesicle Average Intensity caused by this compound during all screening runs. Because the screen was already completed, we could not readily include additional compounds (GNE-3511, ketoconazole). We therefore compared effects of 2BP, **8**, **13**, GNE-3511 and ketoconazole on DLK-GFP puncta in follow-up side-by-side experiments under very similar conditions to the initial screen. Results of these experiments, shown in a new Fig S7, confirm that **8**, **13** and ketoconazole reduce DLK-GFP puncta to a similar extent, but that none are as effective as 2BP. Unexpectedly, GNE-3511 reduced DLK-GFP puncta size/intensity in some cells but caused large accumulations of DLK-GFP in others. Neither of these types of DLK-GFP distribution fell within our defined cut-offs for 'DLK-GFP puncta', so quantified data in Fig S7 thus show that GNE-3511 also reduces DLK-GFP puncta number. While there are clearly additional mechanisms at play in GNE-3511-treated HEK293T cells, we hope the reviewer would agree that defining this effect is better suited to a follow-up study.*

2. Providing a flowchart illustrating high-content screening and analyses, with details of selection criteria.

We have now added a new flowchart with details of selection criteria for specific steps as requested (new Fig S8). Additional information is provided in Table S2.

3. Highlighting control groups and the identified hit compounds in high-content screening results.

*We have added a line on Fig 3B, 3C to show the average effect of 2BP, which was included in all screening plates. We have also added a new Fig S4A, B highlighting the positions of the 33 hit compounds (4500 compound bins, to avoid issues of 'dot density' obscuring hits). For 2BP, GNE-3511, ketoconazole, **8** and **13** we now show representative images and quantification of effects on DLK-GFP puncta in follow-up assays (new Fig S7).*

4. Clearly describing the neurodegeneration index calculation method.

We now provide a more detailed methodology for our new, more quantitative analysis of cell body viability (Methods and new Fig 5). Specifically, we write:

“Assays of DRG neuron cell body viability and axon integrity

After performing TD at DIV5 as above, DRG cultures were returned to the incubator for 45h. Medium was then aspirated, and neurons were incubated for 5 minutes at 37°C with Calcein-AM UltraPure grade (CAS 148504-34-1; 1 mg/ml in DMSO) at a 1:1000 dilution in Neurobasal (NB) medium. Calcein-AM-containing medium was then aspirated, and neurons were subsequently incubated with Hoechst 33342 (10 mg/ml in dH₂O stock) at a 1:2000 dilution (8 µM final concentration) in 1× PBS for 5 minutes at 37°C. Phase contrast, and blue and green channel fluorescent images (Hoechst 33342, Calcein-AM fluorescence, respectively) were then acquired using an EVOS M5000 microscope (20× objective, 0.45NA) in an unbiased manner by imaging the center of the field and then moving to capture five random, non-overlapping fields per well. Calcein-AM images were auto-thresholded using the “Default” mode in ImageJ/Fiji. Hoechst images were auto-thresholded using the “Moments” mode. A mask of overlapping Calcein-AM and Hoechst signals was created using the Image Calculator's "AND" function to identify and count double-positive (i.e. viable) cells. The "Analyze Particles" algorithm was applied to identify these viable cells, using size parameters of 500 to infinity pixels and circularity of 0.8 - 1.0.”

5. Optimizing in vitro neurodegeneration assays for more pronounced degeneration.

We hope the reviewer would agree that the 5-fold change in neuronal viability, as measured using our new assay (new Fig 5A, B), is more than adequate to assess the extent of neurodegeneration and protection.

6. Including quantitative analyses on specific neurodegeneration features, such as axon fragmentation or neurite length, and use more sensitive methods, e.g., fluorescent stainings of specific biomarkers, to assess neurodegeneration in the in vitro assays.

*We appreciate this point. In addition to our new assay of cell body degeneration (summarized in point 5 above), we also developed a new assay to specifically quantify axon integrity. Because axons were not fully degenerated at the time point that gave optimal dynamic range in our assay of cell body viability, we used the ImageJ/Fiji Interactive Watershed function to identify long sections of continuous i.e. unblebbed, phenotypically intact axons. Results from this assay largely mirrored those seen with the cell body viability assay i.e. compounds **8** and **13** were the most axo-protective. These data are shown and quantified in a revised Fig 5A and 5C and example images of the long unbroken axon structures that are quantified are shown in a new Fig S6.*

7. Clarifying the inconsistency in MKK4 phosphorylation data.

We have replaced the original phospho- and total MKK4 blots in Fig 6E with panels that more accurately reflect the pooled, quantified data.

8. Providing a more detailed characterization of compounds 8 and 13, including off-target effects and dose-response relationships.

*We now provide dose-response data, including IC50 values calculated from 8-point dilution curves, for both **8** and **13** in our assay of TD-induced cell body viability (new Fig S9A-D). We also now show that our compounds do not affect palmitoylation of three additional control palmitoyl-proteins in axons following TD (new Fig S12). We also show that our compounds do not affect basal palmitoylation of DLK (new Fig S10) or basal levels of pMKK4 (new Fig S11). Furthermore, we show that our compounds do not prevent other signaling events triggered by NGF withdrawal (shutdown of ERK and Akt pathways; new Fig S13).*

9. Exploring mechanisms by which compounds 8 and 13 specifically inhibit stimulus-dependent DLK palmitoylation.

*We appreciate the reviewer raising this point. Precise mechanism of action is often challenging to define for compounds identified in cell-based screen, but additional experiments performed in response to review have provided several key insights for **8** and **13**. For example, we now show that neither **8** nor **13** prevents TD-induced shutdown of the pro-survival ERK and Akt pathways (new Fig S13). One key conclusion from these results is that neither **8** nor **13** broadly blocks all effects of TD. Moreover, because Akt inhibition/dephosphorylation is sufficient to phenocopy the effect of TD and drive DLK-dependent degeneration in DRG neurons (PMID: 26898330), the neuroprotective action of **8** and **13** is thus more likely due to*

*their action on a step downstream of Akt shutdown. This model is again consistent with the notion that these compounds' key MOA involves their regulation of axonal DLK palmitoylation. As discussed above (Point #2) we also show that neither **8** nor **13** reduces palmitoylation of multiple other axonal palmitoyl-proteins (new Fig S12) i.e. these compounds do not broadly block palmitoylation or stimulate depalmitoylation. We further show that neither **8** nor **13** reduces basal DLK palmitoylation, or pMKK4 levels in the continued presence of NGF (new Figs S10, S11). Finally, we now show that **8** prevents DLK-dependent signaling in response to multiple additional stimuli (nocodazole, axotomy) but that **13** does not block axotomy-induced c-Jun phosphorylation (new Fig S14). These findings are consistent with a model in which **8** broadly blocks stimulus-dependent DLK palmitoylation and signaling, perhaps by inhibiting the protein acyltransferase for DLK. In contrast, **13**, which more selectively prevents certain forms of palmitoyl-DLK signaling, may prevent a specific trafficking step or other cellular event, upstream of DLK palmitoylation and vesicle recruitment. This event would be common to TD- and nocodazole-induced signaling but would not occur, or can be circumvented, after axotomy. Importantly, the likely different points of action for **8** and **13** suggest that there are at least two proteins/processes that could be targeted therapeutically to block palmitoyl-DLK-dependent signaling. We have added a paragraph summarizing the above points to our revised Discussion. We hope the reviewer would agree that these new findings add substantial mechanistic insight but that fully defining mechanism of action is better left to a follow-up study.*

10. Reporting and analyzing the sex of animals used in all experiments.

We have included information on the sex of mice used as requested. Specifically, we write:

“Six-week-old mice of both sexes (14 male, 13 female, assigned randomly across conditions) were used for this study...

...In accordance with Nature Communications guidelines, no post hoc sex- and gender-based analysis was performed due to low sample size when results were segregated by sex.”.

However, because Nature Communications discourages conducting sex-based post hoc analysis of data if sample size is low, we did not segregate results by sex. We nonetheless note that the sample sizes used for our study (n=3-7 mice per condition) are similar to, and in some cases greater than, those used in similar studies of DLK-dependent responses to ONC e.g. PMID: 23431164; PMID: 23431148; PMID: 24878510

11. Including the images of sham controls for all in vivo experiments.

We have modified Figure 8 to include images of sham controls as requested.

12. Expanding in vivo experiments to include RGC survival, nerve fiber integrity, DLK localization, and functional assessments.

We appreciate this point and, as requested and discussed above, began by assessing the effect of 2BP on ONC responses. However, as mentioned above, we found that 2BP increased c-Jun phosphorylation in non-RGC cells in the retina, even after sham injury. While we recognize the potential importance of other readouts suggested (nerve fiber integrity, functional measurements), as discussed above, these processes are only impacted several days-to-weeks post-ONC (e.g. PMID: 33207199; PMID: 23431164; PMID: 23431148). However, the rapid clearance of small molecules from the retina (PMID: 37657528; PMID: 31430156) means that assessing these readouts would require repeated intravitreal injections of our compounds (perhaps once to twice daily for several days). We hope the reviewer would agree that the high potential for false-negative results and/or complications of interpretation due to technical issues makes these experiments best suited to a future study. However, we do discuss this issue, and suggest that controlled release strategies using intravitreal implants, could be used to assess these downstream sequelae and may even be a possible therapeutic approach, albeit one that is beyond the scope of the current manuscript.

[Clarity and context]

The manuscript is generally well-written with appropriate context. Improvements could include:

1. A more detailed explanation of the high-content imaging screen methodology, especially the normalization and handling of plate-to-plate variability.

An excellent point. Our new Figure S3A, B shows Z-factor values (a measure of assay robustness, with Z-factor >0.5 classically considered an excellent assay; PMID: 10838414) for both readouts (puncta per NLS and Vesicle Average Intensity) for initial test runs using the scaled-up conditions for the current screen (compared with Martin et al., 2019). Although our assay platform can achieve a Z-factor approaching or >0.5 for these readouts, in practice we did not achieve this value for all plates in the main screen (plotted in a new Fig S3C, D). We nonetheless consider that the median Z-factor values across the entire screen (P/NLS: 0.41; VAI: 0.49) and our consideration of both readouts when selecting compounds for follow-up is

more than acceptable for a cell-based screen, particularly given that it is also now appreciated that a Z-factor value of >0.5 is somewhat arbitrary, and can be challenging to achieve in cell-based assays, and that assays with Z-factor values <0.5 can still identify useful compounds PMID: 32749188. Importantly, our scaled-up assay indeed successfully identified compounds that reduced both readouts in our primary screen, and which were then highly active in orthogonal assays in primary neurons.

2. A flowchart of high-content screening and analyses with key decision points and criteria.

We have added a new flowchart with details of selection criteria for specific steps as requested (new Fig S8). Further information is provided in a new Table S2.

3. Clearer highlighting of neurodegenerative features of axonal blebbing/beading and cell body degeneration in representative images.

*We appreciate this point. In addition to our new assay of cell body degeneration (summarized in point 5 above), we also developed a new assay to specifically quantify axon integrity. Because axons were not fully degenerated at the time point that gave optimal dynamic range in our assay of cell body viability, we used the ImageJ/Fiji Interactive Watershed function to identify long sections of continuous i.e. unblebbed, phenotypically intact axons. Results from this new assay largely mirrored those seen with the cell body viability assay i.e. compounds **8** and **13** were the most axo-protective. These data are shown and quantified in a revised Fig 5A and 5C and example images showing the intact axons that are counted in this readout are shown in Fig S6.*

4. A schematic diagram illustrating the proposed working model of DLK regulation by palmitoylation and compound effects.

We have added a schematic (new Fig S15) to illustrate our proposed working model, contrasting the effects of DLKi-3511 and our novel compounds.

[References]

The referencing is appropriate, citing key studies on DLK function, its role in neurodegeneration, and previous attempts at therapeutic targeting. They also properly reference their prior work on DLK palmitoylation and high-content screening platforms.

We thank the reviewer for this positive evaluation.

[Reviewer expertise]

The manuscript's primary focus on neurodegeneration, DLK signaling, and pharmacological screening is well within the reviewer's expertise. While the reviewer has a general understanding of pharmacokinetics, the specific methodologies and detailed analysis of these aspects might be better evaluated by experts in those fields.